# Heterotrimeric Gq proteins act as a switch for GRK5/6 selectivity underlying β-arrestin transducer bias

Kouki Kawakami[1,6], Masataka Yanagawa [2,6], Suzune Hiratsuka[1], Misaki Yoshida[1], Yuki Ono[1], Michio Hiroshima[2,3], Masahiro Ueda [3,4], Junken Aoki[5], Yasushi Sako [2✉] & Asuka Inoue [1✉]

Signaling-biased ligands acting on G-protein-coupled receptors (GPCRs) differentially activate heterotrimeric G proteins and β-arrestins. Although a wealth of structural knowledge about signaling bias at the GPCR level exists (preferential engagement of a specific transducer), little is known about the bias at the transducer level (different functions mediated by a single transducer), partly due to a poor understanding of GPCR kinase (GRK)-mediated GPCR phosphorylation. Here, we reveal a unique role of the Gq heterotrimer as a determinant for GRK-subtype selectivity that regulates subsequent β-arrestin conformation and function. Using the angiotensin II (Ang II) type-1 receptor (AT1R), we show that β-arrestin recruitment depends on both GRK2/3 and GRK5/6 upon binding of Ang II, but solely on GRK5/6 upon binding of the β-arrestin-biased ligand TRV027. With pharmacological inhibition or genetic loss of Gq, GRK-subtype selectivity and β-arrestin functionality by Ang II is shifted to those of TRV027. Single-molecule imaging identifies relocation of AT1R and GRK5, but not GRK2, to an immobile phase under the Gq-inactive, AT1R-stimulated conditions. These findings uncover a previously unappreciated Gq-regulated mechanism that encodes GRK-subtype selectivity and imparts distinct phosphorylation-barcodes directing downstream β-arrestin functions.

[1] Molecular and Cellular Biochemistry, Graduate School of Pharmaceutical Sciences, Tohoku University, 6-3, Aoba, Aramaki, Aoba-ku, Sendai, Miyagi 980-8578, Japan. [2] Cellular Informatics Laboratory, RIKEN Cluster for Pioneering Research, 2-1 Hirosawa, Wako, Saitama 351-0198, Japan. [3] Laboratory for Cell Signaling Dynamics, RIKEN BDR, 6-2-3, Furuedai, Suita, Osaka 565-0874, Japan. [4] Laboratory of Single Molecule Biology, Graduate School of Frontier Biosciences, Osaka University, 1-3 Yamadaoka, Suita, Osaka 565-0871, Japan. [5] Graduate School of Pharmaceutical Sciences, The University of Tokyo, 7-3-1, Hongo, Bunkyo-ku, Tokyo 113-0033, Japan. [6]These authors contributed equally: Kouki Kawakami, Masataka Yanagawa. ✉email: sako@riken.jp; iaska@tohoku.ac.jp

GPCRs are the largest family of membrane proteins and are well-established pharmaceutical targets that play a key role in a broad spectrum of physiological events[1]. Agonist stimulation at GPCRs activates heterotrimeric G proteins and their effector proteins, leading to a series of cellular responses. Agonist-bound GPCRs undergo phosphorylation in their cytosolic loops and/or the C-terminal tail by GPCR kinases (GRKs) and subsequently recruit β-arrestin[2]. β-arrestin terminates G-protein-mediated signaling by spatially competing with G protein and facilitating receptor internalization[3,4]. Besides, β-arrestin induces its own signaling by scaffolding effector proteins[5].

Biased ligands have attracted much attention owing to its potential to discriminate a therapeutic effect from other on-target adverse effects[6]. Biased agonism, also known as functional selectivity, is divided into "GPCR bias" and "transducer bias"[7]. GPCR bias is defined as different activation states of GPCRs that allow for binding of a specific transducer (e.g., G protein and β-arrestin). Transducer bias arises from variable conformations of a single transducer (typically, β-arrestin) that results in distinct downstream signaling events. Unlike the well-characterized GPCR bias[8,9], β-arrestin transducer bias remains poorly explored and is proposed to occur through at least two mechanisms: 1) phosphorylation patterns of receptors, 2) β-arrestin-binding modes to receptors.

The former mechanism of the transducer bias is termed a phosphorylation barcode and is produced by a specific pattern of GRK-mediated receptor phosphorylation[10]. In humans, GRKs consist of seven subtypes (GRK1–7) with four subtypes (GRK2, GRK3, GRK5 and GRK6, classified into the GRK2/3 subfamily and the GRK5/6 subfamily) being widely expressed in the body[11]. Biased ligands induce the recruitment of distinct GRK subtypes to the receptor as compared with balanced ligands. It has been reported that individual GRKs recognize different GPCR conformations[12,13]. Further, previous studies identified several mechanisms following G-protein activation that regulate GRK-mediated GPCR phosphorylation. For example, in GRK2/3, binding to a free, membrane-anchored Gβγ subunit plays a critical role in receptor phosphorylation by recruiting GRK2/3 to the plasma membrane from the cytosol and enhancing kinase activity[14]. In GRK5/6, calmodulin-binding negatively regulates receptor phosphorylation by dispersing their localization from the plasma membrane[15]. In addition, certain phospholipids (e.g., phosphoinositides) enhance GRK5/6 kinase activity[16]. GRKs, in turn, can regulate a function of their effector proteins. In GRK2/3, binding to an GTP-bound, activated Gαq subunit inhibits GTPase activity[17,18], while kinase activity of GRK2/3 is not altered[18]. From these findings, it is assumed that G-protein-less-active GPCR agonists rely on GRK5/6, but not GRK2/3, for their β-arrestin recruitment, yet without such evidence or existence of a possible positive regulator for GRK5/6.

The latter mechanism of the transducer bias is visualized in recent structural studies[19,20] that provide two unique engagement modes of GPCR-β-arrestin complexes: tail hanging and core engagement. The tail-hanging form was initially considered as an intermediate state to the core-engagement form, but recently it has been shown that both β-arrestin forms are capable of inducing GPCR endocytosis and Src activation[21]. Despite the differential engagements of β-arrestin to the receptor, their functional distinction is poorly characterized.

Multicolor single-molecule-tracking (SMT) analysis is a powerful method for analyzing dynamic interaction between GPCRs and their effector proteins. A previous SMT study observed accumulation of ligand-bound GPCR molecules in a compartmentalized region in the plasma membrane called "hotspot", where the GPCR molecules encounter a heterotrimeric G protein[22]. While GPCR-triggered G-protein signal transduction

occurs in the hotspot, it is unclear where, i.e. within or outside of the hotspot, subsequent signaling events such as GRK-mediated GPCR phosphorylation and β-arrestin recruitment take place.

AT1R is a widely studied model for biased agonism. TRV027 was developed as a β-arrestin-biased AT1R ligand that induces preferential engagement of Gq over β-arrestin and is tested in a clinical trial for the treatment of heart failure and, more recently, COVID-19-associated acute respiratory distress syndrome[23–25]. A series of in vitro structural studies and a cell-based assay have demonstrated a conformational landscape of AT1R that is dependent on its bound ligands[26–29], as well as the conformation of β-arrestin as a transducer[30]. However, these analyses have again focused on GPCR bias and the relative impact of transducer bias at β-arrestin remains unclear.

Here, we use a battery of complementary approaches including GRK-deficient HEK293 cell lines, split-luciferase-based proximity assays and high-resolution single-molecule-tracking analysis to demonstrate that, with reference to Ang II, TRV027 induces β-arrestin transducer bias by selectively enhancing GRK5/6-dependent phosphorylation in AT1R. We surprisingly discover that active GRK5/6 availability for ligand-bound AT1R is negatively regulated by the activity of a G protein, i.e., Gq, which has typically been considered a non-factor in the context of β-arrestin bias at GPCRs.

## Results

**Selectivity of GRK5/6 at AT1R inversely correlates with Gq activity.** Previous studies indicate that phosphorylation of AT1R is mediated by GRKs[31] and members of the protein kinase C (PKC) family[32]. To address whether these kinases are exclusively responsible for β-arrestin recruitment to ligand-bound AT1R, we genetically deleted GRKs and treated the cells with a pan-PKC inhibitor Go6983. Given that the ubiquitously expressed four GRK members (GRK2, 3, 5, 6) are grouped into GRK2/3 and GRK5/6[33], we generated HEK293 cell lines lacking GRK2/3, GRK5/6 or combination of the two (referred to as ΔGRK2/3, ΔGRK5/6 or ΔGRK2/3/5/6 cell lines, respectively) by using a CRISPR/Cas9 technology (Fig. 1a). Western blot analysis confirmed that the targeted GRKs were successfully eliminated (Fig. 1b, c). Expressions of untargeted GRKs were increased by two-to-three fold and the altered GRK levels correlated with the variability of the β-arrestin-recruitment response (below) between the clones for the same genotypes (Supplementary Fig. 1, 2a–d). Furthermore, expression and functional analyses of GPCR signal components identified variability among the genotypes and the clones, thus suggesting a need for a cautious interpretation when comparing GPCR signaling responses among the GRK-deficient cell lines (Supplementary Fig. 2e–l). Nevertheless, we viewed that the GRK-deficient cells would be valuable tools to dissect bias at the level of GRK subtypes and that such clone-associated issues would be minimized when GRK subtypes are re-expressed in ΔGRK2/3/5/6 (below). Using the NanoBiT-based protein-protein interaction assay[34], we measured the association of β-arrestin1 and β-arrestin2 with AT1R in the presence of Go6983 (hereafter included for pretreatment unless otherwise noted). Ang II-induced β-arrestin1 recruitment to AT1R was abolished in ΔGRK2/3/5/6 cells while the responses were partially reduced in ΔGRK2/3 cells and rather increased in ΔGRK5/6 cells (Fig. 1d, e), the latter of which may be in part attributable to the upregulated GRK2/3 expression. Notably, in the absence of Go6983, we observed essentially the same relative contributions of the GRK subtypes as the Go6983-treated condition (Supplementary Fig. 3a, b), excluding a direct effect of PKC inhibition on GRK regulation[35–37]. β-arrestin2 showed a similar GRK dependency to that of β-arrestin1 although there was a residual recruitment

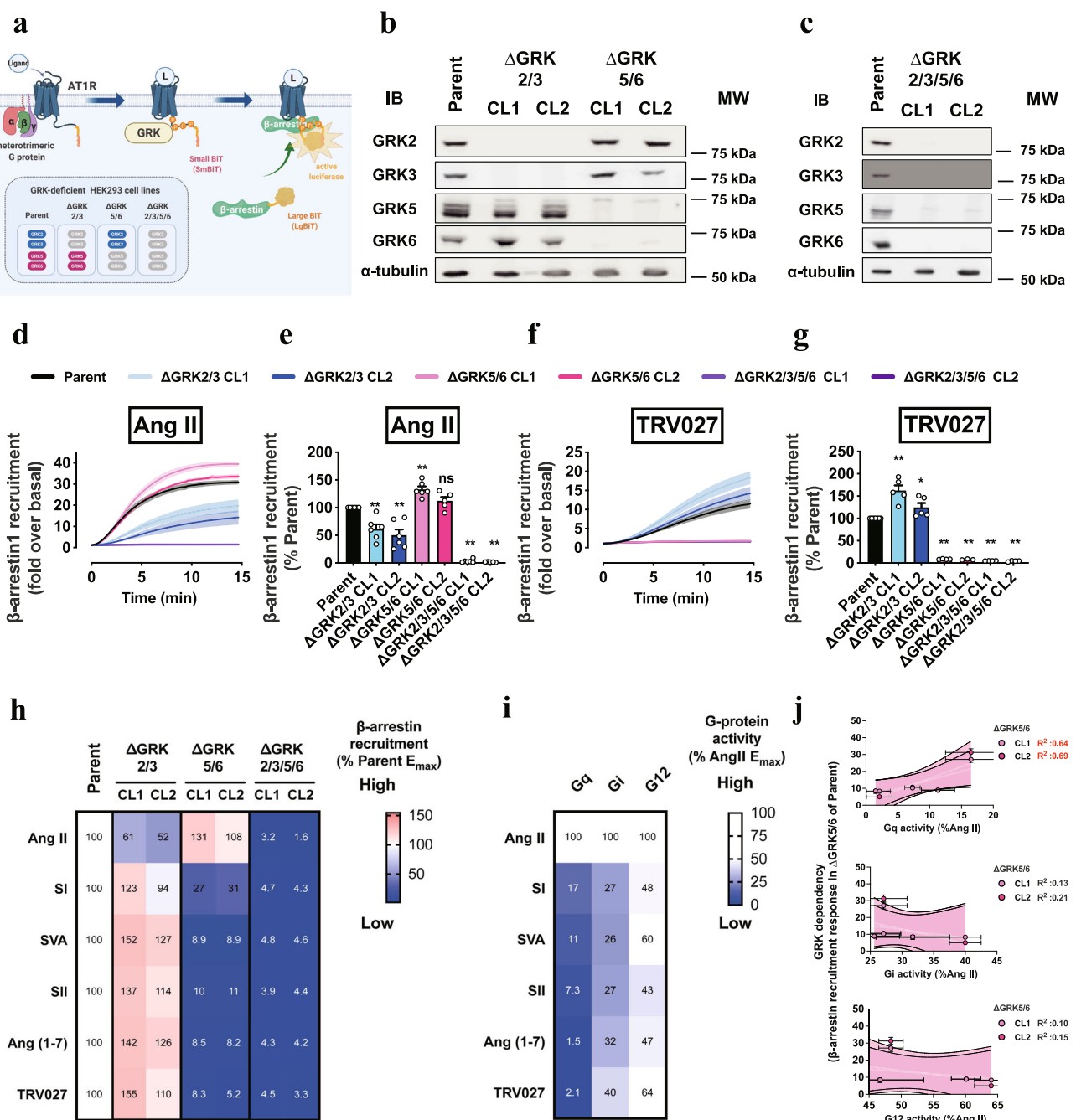

**Fig. 1 GRK-subtype selectivity switch by β-arrestin-biased AT1R ligands. a**, Schematic representation of the measurement of GRK-subtype selectivity. The NanoBiT pair consisting of SmBiT-fused AT1R and LgBiT-fused β-arrestin was co-expressed in the parent or the indicated GRK-deficient cells and AT1R ligand-induced β-arrestin recruitment was monitored. **b**, **c**, Validation of successful elimination of the targeted GRK subtypes. Western blot analyses were performed by using lysates derived from wild-type (parent HEK293) and ΔGRK2/3, ΔGRK5/6 (b) and ΔGRK2/3/5/6 (c). Membranes were probed with the indicated antibodies against GRK2, 3, 5, 6 or α-tubulin. α-tubulin was used as a loading control. **d**–**g**, Luminescent kinetics (d, f) and quantifications (e, g) of the NanoBiT-β-arrestin-recruitment responses. The indicated cells were stimulated with 1 μM Ang II (d, e) or 1 μM TRV027 (f, g). **h**, Heatmap representation of the β-arrestin recruitment by Ang II and its analogs. Data are presented as mean from 3–6 independent experiments and β-arrestin-recruitment signal in the parent cells was set as 100% for each ligand. **i**, G-protein-coupling signatures (derived and re-edited from Namkung et al.[25]). **j**, Comparison of the β-arrestin recruitment responses in ΔGRK5/6 cells and G-protein-coupling activity. The scattered plots represent data from the five ligands except for Ang II (see also Supplementary Fig. 3i, j for plots with all of the cell lines and the six ligands). In Fig. 1b, c, representative images of 3–4 independent experiments are shown. In Fig. 1d, f, lines and shaded regions represent mean and SEM, respectively, of 3–6 independent experiments with each performed in duplicate. In Fig. 1e, g, bars and error bars represent mean and SEM, respectively, of 3–6 independent experiments with each performed in duplicate. In Fig. 1j, symbols and error bars represent mean and SEM, respectively, of 3–6 independent experiments with each performed in duplicate. Shaded regions denote 90% confidence of the linear regression analysis. $R^2$ values are shown at the right. Note that for many data points, vertical error bars are smaller than the size of the symbols, and thus are not visible. In Fig. 1e, g, * and ** represent $P < 0.05$ and 0.01, respectively, with one-way ANOVA followed by the Dunnett's test for multiple comparison analysis with reference to the Ang II stimulation. ns, not significantly different between the groups. See Supplementary statistics data file for additional statistics and exact $P$ values.

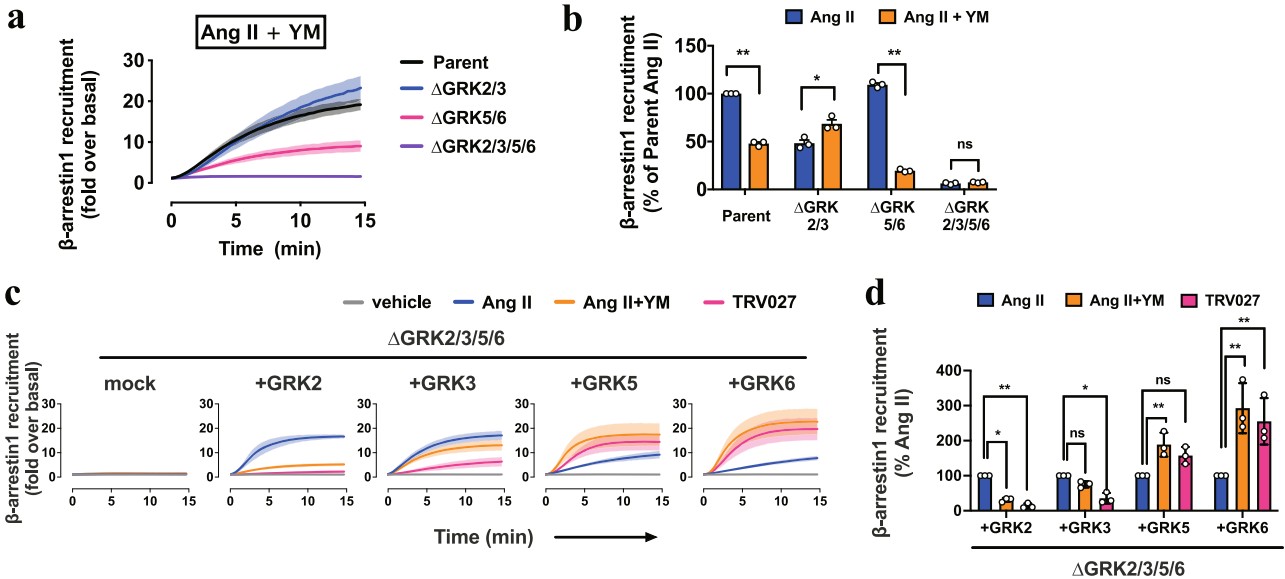

**Fig. 2 Gq inhibition in Ang II mimics the GRK-subtype selectivity of TRV027. a**, Luminescent kinetics of β-arrestin1-recruitment responses upon 1 μM Ang II stimulation in the presence of pretreatment of the Gq inhibitor, YM-254890 (1 μM). **b**, Comparison of Ang II-induced β-arrestin1 recruitment in the parent and the GRK-deficient cell lines in the presence or absence of YM-254890. For each experiment, Ang II-induced β-arrestin1-recruitment signal in the parent cells was set as 100%. **c**, **d**, Assessment of the individual GRK subtypes for their sensitivity to YM-254890. The C-terminally FLAG epitope-tagged GRK2, 3, 5 and 6 constructs were individually expressed at a level equivalent to that of endogenous GRK2 (Supplementary Fig. 3a, b) along with AT1R-Sm and Lg-β-arrestin1 in ΔGRK2/3/5/6 cells and stimulated with 1 μM Ang II. For each experiment, Ang II-induced β-arrestin1-recruitment signal was set as 100% (d). Cell lines used were ΔGRK2/3, CL2; ΔGRK5/6, CL2; ΔGRK2/3/5/6, CL2 (a–d). In Fig. 2a, c, lines and shaded regions represent mean and SEM, respectively, of 3 independent experiments with each performed in duplicate. In Fig. 2b, d, bars and error bars represent mean and SEM, respectively, of 3 independent experiments with each performed in duplicate. In Fig. 2b, d, * and ** represent $P < 0.05$ and $0.01$, respectively, with two-way ANOVA followed by the Dunnett's test for multiple comparison analysis with reference to the Ang II stimulation. ns, not significantly different between the groups. See Supplementary statistics data file for additional statistics and exact $P$ values.

response even in ΔGRK2/3/5/6 cells (Supplementary Fig. 3c, d). These results indicate that Ang II-stimulated β-arrestin1 and β-arrestin2 recruitment is primarily mediated by GRK2/3 (as in ΔGRK5/6 cells) with a minor contribution from GRK5/6 (as in ΔGRK2/3 cells) and that there are no other kinases involved in β-arrestin1 recruitment except for PKCs (Supplementary Fig. 3e, f).

We performed the same experiment using biased ligands for AT1R including TRV027. Intriguingly, β-arrestin recruitment was greatly attenuated in ΔGRK5/6 cells and rather enhanced in ΔGRK2/3 cells (Fig. 1f, g). We additionally assessed four β-arrestin-biased AT1R ligands with different Gq signaling activities (SI, SVA, SII and Ang (1–7)) and illustrated their GRK dependency (i.e., reduction of β-arrestin-recruitment response in the GRK-deficient cell lines) as a heatmap (Fig. 1h and Supplementary Fig. 3g, h). Comparison of the GRK-subtype selectivity with a previously reported G-protein-coupling profile[38] indicates that GRK5/6-dependent β-arrestin-recruitment response is inversely correlated with Gq activity, but not Gi or G12 activity (Fig. 1i, j and Supplementary Fig. 3i, j).

**Inhibition of Gq activation alters GRK-subtype selectivity and β-arrestin function.** Recent studies have shown that Ang II and TRV027 induce unique structural changes in AT1R, suggesting that GPCR bias is a direct consequence of these structural states[29]. Given our finding that the bias exists at the level of GRK-subtype selectivity (Fig. 1h), we questioned whether the lack of Gq activation or the ligand-bound AT1R state is a determinant for the GRK-subtype selectivity. To test this, we pretreated the GRK-deficient cell lines with the chemical Gq inhibitor YM-254890 (YM))[39] and assessed Ang II-induced β-arrestin recruitment. YM pretreatment reduced β-arrestin-recruitment response in ΔGRK5/6 cells, but enhanced the response in

ΔGRK2/3 cells (Fig. 2a, b and Supplementary Fig. 4a, b). To evaluate the contributions of individual GRKs and to exclude a clonal effect of the knockout cell lines, we restored each GRK subtype in ΔGRK2/3/5/6 cells at comparable levels across the GRK subtypes. By quantitative real-time PCR analyses, we observed that mRNA expression levels of the GRK subtypes was highest in GRK2, followed by GRK6, GRK3 and GRK5 (Supplementary Fig. 2g). Using C-terminally FLAG epitope-tagged GRK recombinant proteins and antibodies for FLAG tag and the individual GRK subtypes, we found that GRK2 was expressed about one order higher than the other GRK subtypes (Supplementary Fig. 4c, d). In the following add-back experiments, we individually restored the GRK subtypes in ΔGRK2/3/5/6 cells at an expression level comparable to the endogenous expression of GRK2. Restoration of GRK5 or GRK6, but not GRK2 or GRK3, gained β-arrestin-recruitment signal upon YM pretreatment (Fig. 2c, d and Supplementary Fig. 4e, f). Reduction of GRK2/3-mediated β-arrestin-recruitment responses in the YM pretreatment condition was likely attributable to reduced level of free Gβγ subunits (Supplementary Fig. 4g,h), which triggers GRK2/3-mediated GPCR phosphorylation[27]. In addition, kinase-inactive mutants failed to restore β-arrestin recruitment response (Supplementary Fig. 5a–d), indicating that the observed β-arrestin recruitment thus far is dependent on a kinase activity of GRKs. Together, these data demonstrate that inhibiting Gq activation enhances GRK5/6-dependent β-arrestin recruitment to AT1R.

To identify AT1R residue(s) responsible for the GRK5/6-dependent β-arrestin recruitment, we individually mutated the seven Ser and Thr residues to Ala within the two potential phosphorylation codes (Supplementary Fig. 5e)[28]. These single-Ala mutants were similarly expressed to the level of WT AT1R (Supplementary Fig. 5f), but showed variable levels of ligand-

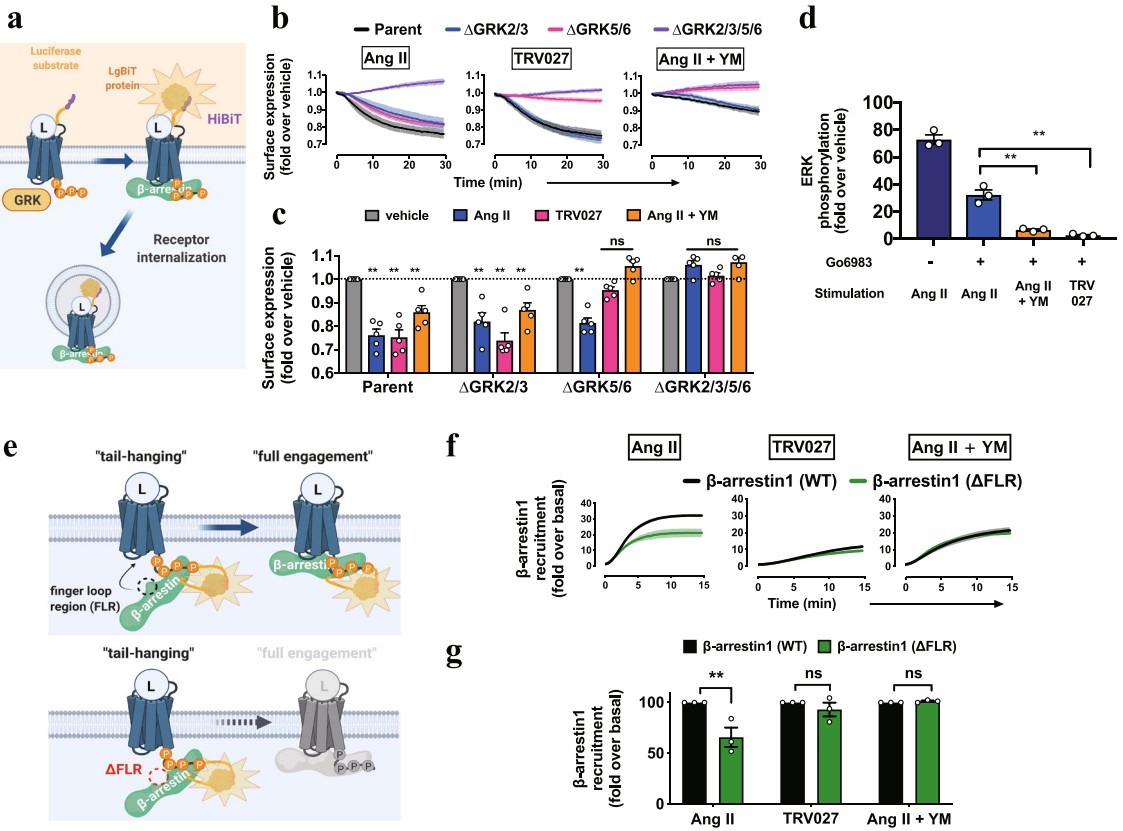

**Fig. 3 GRK-subtype-dependent β-arrestin functionalities. a**, Schematic representation of the HiBiT-based AT1R-internalization assay. N-terminally HiBiT-fused AT1R was expressed in the parent or the GRK-deficient cell lines and a cell-surface AT1R level upon ligand stimulation was measured by complementation with the recombinant LgBiT protein added to the conditioned media. **b**, Luminescent kinetics of the AT1R-internalization assay. The indicated cells were stimulated with 10 nM Ang II, 100 nM TRV027 or 10 nM Ang II pretreated with 1 μM YM-254890. At every measurement time point, a ligand-stimulated count was normalized to a vehicle count. **c**, Quantifications of the AT1R-internalization assay. Bars and error bars represent mean and SEM, respectively, of 5 independent experiments with each performed in duplicate. **d**, AlphaLISA-based pERK measurement. AT1R-expressing cells were pretreated with or without the PKC inhibitor, Go6983 (3 μM), and stimulated with the indicated ligand for 30 min. **e–g**, Schematic illustration (e), luminescent kinetics (f) and quantification (g) of the tail-hanging β-arrestin-recruitment assay. AT1R-Sm and Lg-β-arrestin1 with the truncation in the finger-loop region (ΔFLR) were expressed in the parent cells and treated with the indicated conditions. Shaded regions denote SEM ($n = 3$–5; b, f). Cell lines used were ΔGRK2/3, CL2; ΔGRK5/6, CL2; ΔGRK2/3/5/6, CL2 (a–c). In Fig. 3b, f, lines and shaded regions represent mean and SEM, respectively, of 3 independent experiments with each performed in duplicate. In Fig. 3c, d, g, bars and error bars represent mean and SEM, respectively, of 3 independent experiments with each performed in duplicate. In Fig. 3c, d, g, * and ** represent $P < 0.05$ and 0.01, respectively, with two-tailed multiple $t$-test (d) or one-way ANOVA (g) or two-way (c) ANOVA followed by the Dunnett's test for multiple comparison analysis (with reference to the vehicle stimulation (c) or the Ang II stimulation (g)). ns, not significantly different between the groups. See Supplementary statistics data file for additional statistics and exact $P$ values.

induced β-arrestin1 recruitment. By normalizing TRV027-induced β-arrestin1 recruitment to that of Ang II, we found that TRV027-induced response was selectively less efficacious in S326A (Supplementary Fig. 5g, h). These single Ala-mutant results indicate that Ser326 is a key residue for TRV027-induced GRK5/6 phosphorylation, yet multiple or stepwise phosphorylation may be potentially important.

The importance of Gq in switching GRK5/6 utilization led us to investigate whether β-arrestin transducer bias in reference to Ang II occurred upon TRV027 stimulation or YM pretreatment followed by Ang II stimulation (TRV/Ang II + YM treatments). In the following experiments, we examined the functions and conformations of β-arrestin.

First, we evaluated GPCR internalization using an AT1R construct with an N-terminal HiBiT[34] (Fig. 3a). In the parental cells, Ang II stimulation gradually decreased the cell-surface AT1R. The internalization response was abolished in ΔGRK2/3/5/6 cells, but remained in ΔGRK2/3 and ΔGRK5/6 cells. Remarkably, the TRV/Ang II + YM treatments failed to cause

the internalization response in ΔGRK5/6 cells. These results indicate that both GRK2/3 and GRK5/6 contribute to Ang II-dependent AT1R internalization, whereas GRK5/6 is selectively used following the TRV/Ang II + YM treatments (Fig. 3b, c and Supplementary Fig. 6).

Second, we evaluated phosphorylation of extracellular signal-regulated kinase (pERK), a prototypical signaling event downstream of β-arrestin activation, by an AlphaLISA assay. To eliminate pERK response by the Gq-PKC pathway[40], the AT1R-expressing cells were treated with Go6983. In this condition, we confirmed that the pERK response was dependent on β-arrestin as the pERK signal was completely diminished in the β-arrestin-deficient (ΔARRB) cells (Supplementary Fig. 7a). Ang II stimulation induced a robust pERK signal, pERK response was marginal by TRV027 stimulation[41,42] and nearly undetectable by YM pretreatment (Fig. 3d and Supplementary Fig. 7b–d). We note that the low level of pERK induced by TRV027 was largely diminished upon Go6983 treatment, indicative of a residual Gq-PKC-mediated signaling component (Supplementary Fig. 7b).

These results indicate that internalization and signaling activities of β-arrestin are separable.

Third, we measured a conformational change of β-arrestin using an Ib30 sensor (Supplementary Fig. 7e), a single-chain antibody that recognizes a specific active form of β-arrestin1[43,44]. The TRV/Ang II + YM treatments apparently decreased Ib30 reactivity to β-arrestin1 as compared with Ang II stimulation (Supplementary Fig. 7f, g). Normalization of the Ib30 and β-arrestin-recruitment responses showed that Ib30 recruitment is proportional to the β-arrestin1 recruitment levels (Supplementary Fig. 7h). Thus we conclude that the Ib30-recognizable active β-arrestin conformation is similarly induced in the TRV/Ang II + YM treatments.

Last, we assessed β-arrestin-binding forms. β-arrestin has at least two distinct modes of engagement with GPCRs: the tail-hanging and the core-engagement states[19]. While both are capable of receptor internalization, functional differences, if any, remain obscure[21,45] To distinguish the two modes, we used β-arrestin constructs lacking the finger-loop region (FLR) (Fig. 3e), a critical motif for core engagement[21]. Both ΔFLR β-arrestin1 and 2 were expressed at the level of the WT counterparts (Supplementary Fig. 8a, b) and showed a reduced recruitment response upon Ang II stimulation (Fig. 3f, g and Supplementary Fig. 8c–f). On the other hand, responses by the TRV/Ang II + YM treatments were comparable between WT and ΔFLR β-arrestins. Furthermore, expression of ΔFLR β-arrestins in ΔARRB cells fully induced a ligand-induced AT1R endocytosis (Supplementary Fig. 8g–j), but minimally restored pERK responses (Supplementary Fig. 8k–m). These data support a model in which Ang II-induced β-arrestins are in both engagement modes, whereas β-arrestins upon the TRV/Ang II + YM treatments mostly exist in the tail-hanging mode.

Curiously, during comparison of the two β-arrestin subtypes, we found a unique feature of β-arrestin2. As stated above, β-arrestin2 recruitment was greatly reduced but, unlike β-arrestin1, still detectable in ΔGRK2/3/5/6 cells (Supplementary Fig. 9a–c). Even in a C-terminally phosphorylation-deficient (Δtail-phos) AT1R construct (Supplementary Fig. 9d, e), β-arrestin2 showed a residual recruitment response. Consistently, C-terminally truncated (thus phosphorylation-deficient) AT1R was shown to be competent of recruiting β-arrestin2, but not β-arrestin1[36]. In both cases, ΔFLR β-arrestin2 was completely unresponsive (Supplementary Fig. 9f–h). Besides the GRK-independent recruitment, both β-arrestin1 and 2 showed identical properties such as AT1R endocytosis, ERK phosphorylation and the GRK subtype preference (Supplementary Figs. 2–4, 6). These results indicate that β-arrestin2 is capable of engaging with AT1R via a core interaction without the need for tail phosphorylation (Supplementary Fig. 3c, d).

Taken together, these results indicate that the GRK-subtype selectivity arises not directly from structural states of the ligand-bound receptor, but from the activation level of Gq and, further, that this mechanism governs β-arrestin functionality.

**AT1R and GRK5 interaction occurs in a membrane domain.** Using SMT analysis, we evaluated the changes in dynamics of AT1R and GRK molecules. We monitored SF650-labeled AT1R and TMR-labeled GRK2 or GRK5 in ΔGRK2/3/5/6 cells for their diffusion, oligomerization and colocalization (Fig. 4a and Supplementary Movies 1, 2). In accordance with previous studies[22,46], the variational Bayesian-hidden Markov model (VB-HMM) analysis classified diffusion of the molecules into four states (immobile, slow, medium, and fast; Fig. 4b and Supplementary Fig. 10, 11). Mean square displacement (MSD)-Δt plot analysis showed that AT1R and GRKs in the immobile and the slow states were restricted to membrane domains with a confinement length of 90–250 nm (Supplementary Fig. 11e, f).

We estimated the distribution of putative oligomer size of the molecules from the intensity histogram of the Gaussian functions (Supplementary Fig. 12). Assuming that both AT1R and the GRKs are monomeric in the fast states, we inferred that these molecules are dimers or oligomers in the immobile state. Similarly, AT1R exists as a mixture of dimers and monomers in the slow and the medium states, and GRKs a mixture of dimers and tetramers in these states. Ang II stimulation statically significantly decreased the putative oligomer size in GRK5 in the immobile state (Supplementary Fig. 12c). The TRV/Ang II + YM treatments did not alter it and an increase of the putative oligomer size in the medium and fast states of GRK5. In contrast, no significant change was observed in GKR2 (Supplementary Fig. 12c). Next, we assessed the frequency and duration of colocalization of AT1R with GRK molecules (Fig. 4c–e and Supplementary Fig. 13). Compared with vehicle treatment, Ang II stimulation significantly decreased the association rate between AT1R and GRK5 (Fig. 4d). In the meantime, the lifetime of AT1R-GRK5 complexes was slightly increased (Fig. 4e). The YM pretreatment reverted the Ang II-induced decrease of association rate to the similar extent of TRV027 stimulation. The TRV027 stimulation statically significantly increased the lifetime of AT1R-GRK5 complexes (Fig. 4e and Supplementary Fig. 13e,f.). These results suggested an enhanced availability of ligand-bound AT1R to active GRK5 in the TRV/Ang II + YM treatments.

Finally, we compared the diffusion behaviors of the molecules and the complex (Fig. 4f, g and Supplementary Fig. 11, 13i). Upon Ang II stimulation, AT1R molecules in the GRK2- or the GRK5-expressing cells increased the proportion of the slower fractions (immobile, slow). On the other hand, the TRV/Ang II + YM treatments increased the level of the slower fractions in the GRK5-expressing, but not the GRK2-expressing cells (Fig. 4f). Ang II, but not TRV/Ang II + YM, stimulation increased GRK2 molecules in the slower fraction whereas the TRV/Ang II + YM treatments, but not Ang II stimulation, elevated GRK5 molecules in the slower fraction (Fig. 4g), indicating that AT1R phosphorylation occurs in the slower fraction for both GRKs. Notably, Ang II stimulation decreased the immobile fraction of GRK5 molecules and concurrently accumulated AT1R molecules in the immobile phase. The altered distribution of GRK5 molecules by the Ang II stimulation (i.e., Ang II-induced decrease in the slower fraction) was also observed in the parent cells but not in the Gq family-deficient (ΔGq) cells (Supplementary Fig. 14)., suggesting that the lack of active Gq, rather than the existence of inactive, is a driver for the altered behaviors in GRK5. Collectively, these data support a model in which activation of Gq promotes dissociation of GRK5 from the AT1R at the immobile domain (Fig. 4h). In the meantime, Gq activation triggers GRK2 recruitment to the immobile domain (Fig. 4i) presumably via the resulting free Gβγ, and frees up the receptor for phosphorylation by the activated GRK2/3.

**Assembly and dissociation of the AT1R-Gq-GRK complex.** To gain insight into how Gq activation regulates GRK5/6 selectivity, we assessed the recruitment of GRKs to AT1R (Fig. 5a). β2AR, which was previously shown to be phosphorylated by the four GRK subtypes[11], exhibited an increase in luminescence, indicative of GRK recruitment to the receptor, for all of the four GRKs upon isoproterenol stimulation (Fig. 5b). In contrast, upon Ang II stimulation, AT1R showed recruitment responses for GRK2 and GRK3, but dissociation responses for GRK5 and GRK6 (Fig. 5b, c and Supplementary Fig. 15a, b). Remarkably, the TRV/Ang II + YM treatments induced GRK5/6 recruitment to AT1R (Fig. 5d, e and Supplementary Fig. 15c, d). GRK2/3 showed recruitment response, but at a lower level as compared with Ang II

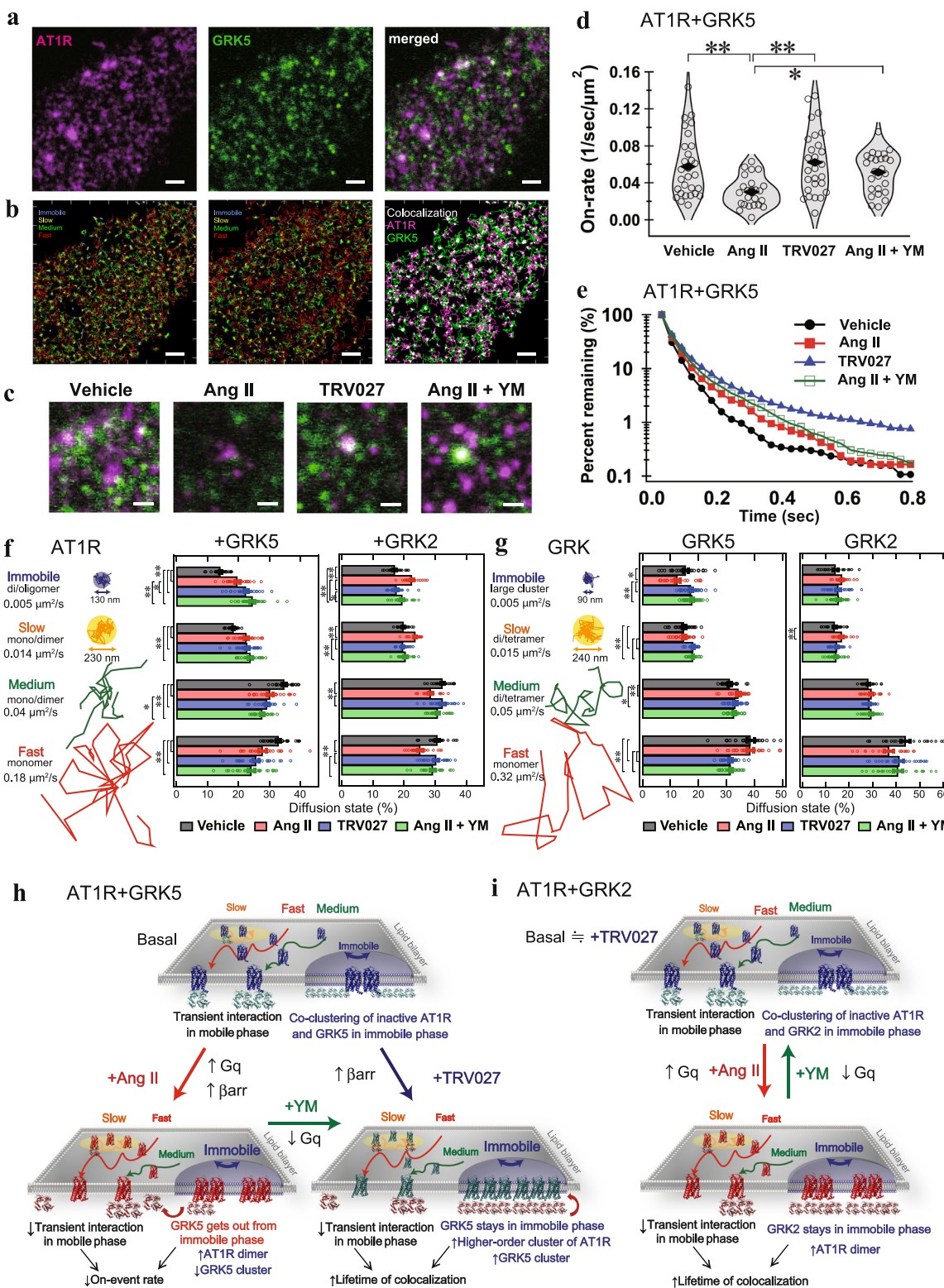

stimulation. These results are consistent with the selectivity of GRK-mediated β-arrestin-recruitment responses by the NanoBiT-based sensors and the single-molecule imaging (Figs. 1d–g, 2a, b, 4).

Next, we assessed whether the presence of inactive Gq or lack of active Gq, or more broadly, the contribution of the other G proteins, is important for the GRK5/6 behavior. To do so, we

conducted the same experiments as above using a panel of the G-protein-deficient cell lines. The GRK5/6 recruitment pattern was reversed in ΔGq cells, but not in the other G-protein-deficient cell lines (Fig. 5f, g and Supplementary Fig. 15e, f). Inhibitors for the Gq signaling pathway (PLCβ, $Ca^{2+}$ and PKC) showed a variable degree of suppression levels, but none of them mimicked the YM-like GRK5/6-association pattern (Supplementary Fig. 16a, b). To

**Fig. 4 Single-molecule imaging reveals accumulation of AT1R and GRK5 in the immobile phase under the Gq-inactivated condition. a**, Representative total internal reflection fluorescence (TIRF) microscopic image of ΔGRK2/3/5/6 (CL2) cells expressing SF650-labeled AT1R (magenta) and TMR-labeled GRK5 (green). Scale bar: 5 μm. **b** Representative trajectories of 4 diffusion states of AT1R (left), GRK5 (middle), AT1R-GRK5 colocalization (right). **c** Representative images of AT1R-GRK5 colocalization. Scale bar: 1 μm. **d**, **e**, AT1R-GRK5 association (on-event) and dissociation (colocalization duration) rates estimated from all trajectories. **f**, **g**, Fractions of the diffusion states in AT1R (f) and GRK5 (g) molecules by the indicated condition (vehicle, 1 μM Ang II, 1 μM TRV027 or 1 μM Ang II pretreated with 1 μM YM-254890). **h**, **i**, Schematic summary of single molecule imaging of AT1R-GRK5 (h) and AT1R-GRK2 (i). In Fig. 4a–c, representative images of 20–29 cells were shown. In Fig. 4d, shaded regions represent a histogram of 20–29 cells. In Fig. 4e, Symbols represent mean of all trajectories in 20–29 cells. In Fig. 4f, g, bars and error bars represent mean and SEM, respectively, of 20–29 cells. In Fig. 4d, e, f, g, * and ** represent $P < 0.05$ and $0.01$, respectively, with one-way ANOVA with followed by the Tukey HSD test among 4 groups ($n = 20$–$29$ cells). See Supplementary statistics data file for additional statistics and exact $P$ values.

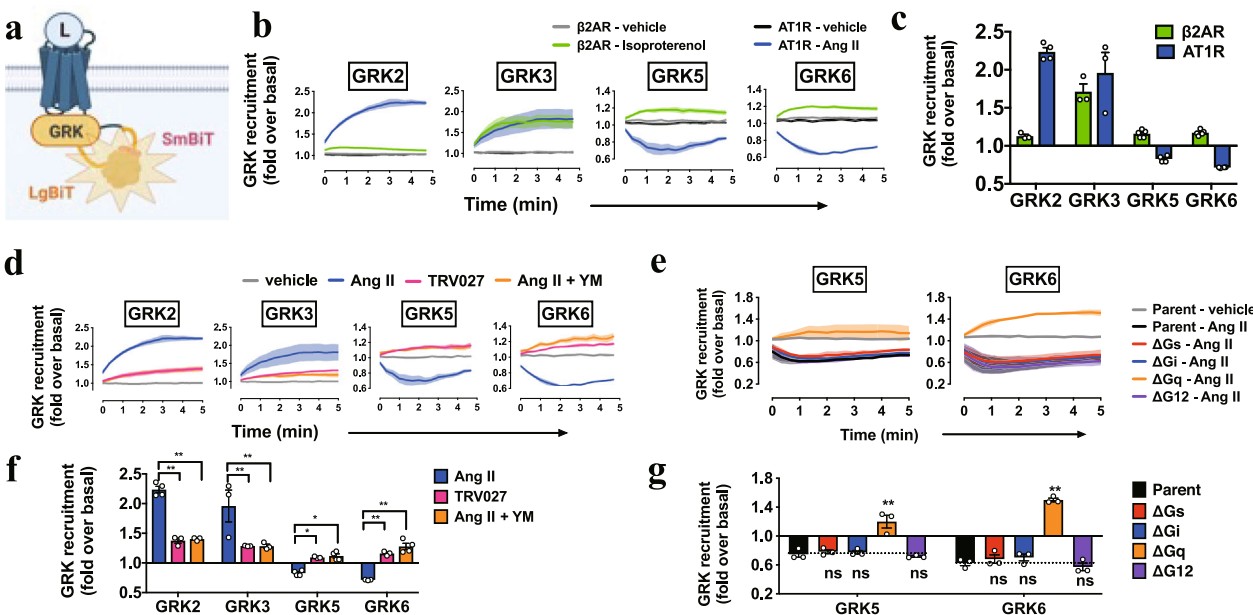

**Fig. 5 Active Gq selectively disperses GRK5/6 from AT1R. a–c**, Schematic representation (a), luminescent kinetics (b), and quantifications (c) of the NanoBiT-GRK-recruitment assay. AT1R-Sm or β2AR-Sm was expressed together with the indicated GRK-Lg constructs in the parent cells and stimulated with Ang II (1 μM) or isoproterenol (10 μM). Note that owing to the time lag between manual ligand addition and start of kinetics measurement, luminescent signals were changed from the basal count (normalized to the value of 1) at the first reading (set as time = 0 min). **d–g**, NanoBiT-GRK recruitment in the TRV/Ang II + YM treatments (d, e) and the G-protein-deficient cells (f, g). HEK293A cells deficient for the individual G-protein families (ΔGs, ΔGi, ΔGq, ΔG12; each deficient for the Gα subunits expressed in HEK293A cells; see Methods for detail) were used to monitor Ang II (1 μM)-induced AT1R-GRK association (f, g). Note that vehicle plots from all of the cell lines overlapped tightly and only those from the parental cells were shown for clarity (f). In Fig. 5b, d, f, lines and shaded regions represent mean and SEM, respectively, of 3 (f) or 3–4 (b, d) independent experiments with each performed in duplicate. In Fig. 5c, e, g, bars and error bars represent mean and SEM, respectively, of 3 (g) or 3–4 (c, e) independent experiments with each performed in duplicate. In Fig. 5e, g, * and ** represent $P < 0.05$ and $0.01$, respectively, with two-way ANOVA followed by the Dunnett's test for multiple comparison analysis (with reference to the Ang II stimulation (e) or the Parent (g)). See Supplementary statistics data file for additional statistics and exact $P$ values.

further validate a specific effect of Gq, we generated cells lacking the Gq family members in the genetic background of ΔGRK2/3/5/6 cells (GRK/Gq-6KO; Supplementary Fig. 16c, d) and assessed GRK5-suppressive activity of the individual Gq members by re-expressing GRK5 along with the NanoBiT-β-arrestin sensor. We found that Ang II-induced β-arrestin-recruitment response in the GRK/Gq-6KO cells was increased to the level of YM-treated ΔGRK2/3/5/6 cells (Supplementary Fig. 16e, f; note that GRK5 was exogenously expressed in both cell lines), validating the on-target effects of YM. Among the four Gq family members, the Gαq, the Gα11 and the Gα14 subunits, but not the Gα16 subunit, exhibited an inhibitory effect on GRK5-mediated β-arrestin-recruitment (note that the experiment was performed in the absence of YM) while all of the members showed equivalent Gq-signaling activity (Supplementary Fig. 16g). Furthermore, the expression of a constitutively active Gαq in the GRK/Gq-6KO cells attenuated the GRK5-mediated β-arrestin recruitment

response whereas such effect was not observed for a dominant-negative Gαq (Supplementary Fig. 16h, i). These data indicate that active Gq, but not downstream Gq signaling, induces dissociation of GRK5/6 from AT1R.

To better understand how Gq activation regulates the GRK5/6 behavior, we evaluated whether GRK5/6 would exist proximal to Gq and this colocalization would be affected by Gq activation states (Fig. 6a, d). We detected significant luminescent signals in cells expressing the Gαq-Lg and the GRK5-Sm or the GRK6-Sm sensor (Supplementary Fig. 17a). Upon Ang II stimulation, both of the Gαq-GRK sensor and the Gβγ-GRK sensor showed decreased luminescence (Fig. 6b, c, e, f and Supplementary Fig. 17b–e). These luminescence changes were suppressed or reversed by the TRV/Ang II + YM treatments, indicating that GRK5/6 dissociates from the Gq heterotrimer in a Gq activation-dependent manner.

Finally, we tested whether the three components (AT1R, Gq and GRK5/6) were in close proximity by using a combined

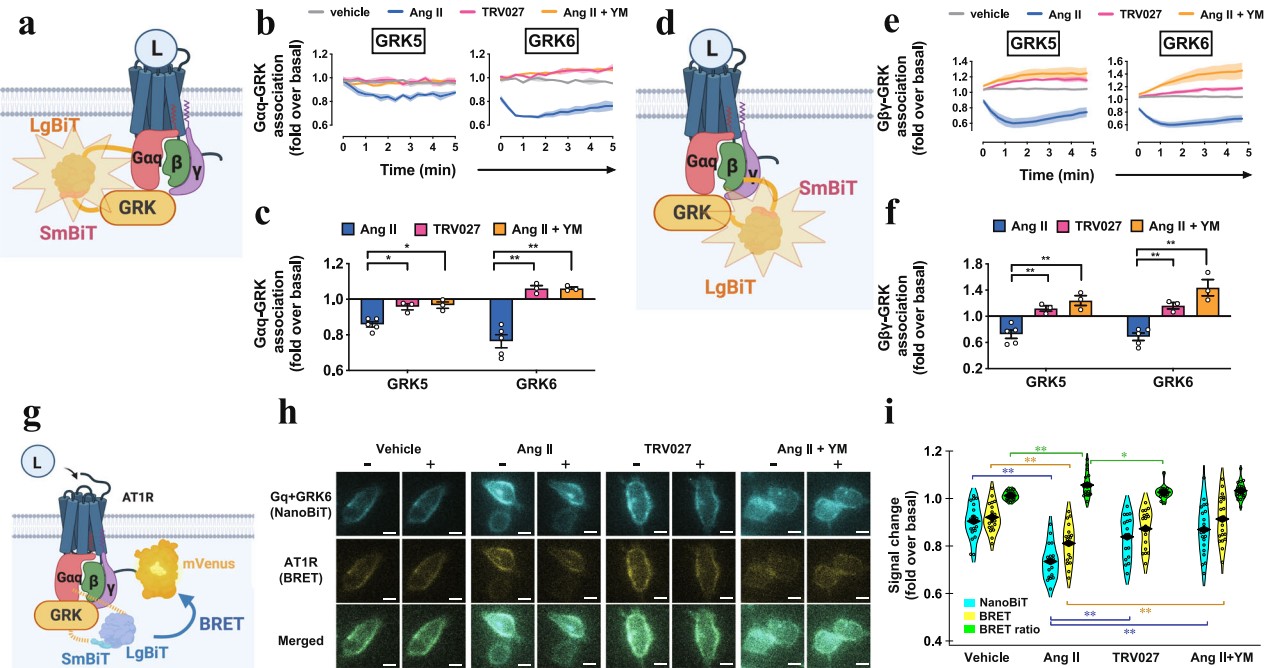

**Fig. 6 Colocalization of AT1R, Gq and GRK5 molecules. a–c**, Schematic representation (a), luminescent kinetics (b), and quantifications (c) of the Gαq-GRK proximity assay. LgBiT-fused Gαq subunit and SmBiT-fused GRK were expressed along with AT1R and the Gβ1 and the Gγ2 subunits in the parent cells. **d–f** Schematic representation (d), luminescent kinetics (e), and quantifications (f) of the Gβγ-GRK proximity assay. Same as in (a–c), but using the SmBiT-fused Gβ1 and the LgBiT-fused GRK constructs. The indicated reporter-expressing cells were stimulated with Ang II (1 μM), TRV027 (1 μM) or Ang II (1 μM) + YM (1 μM) (b, c, e, f). Shaded regions denote SEM ($n = 3$–$5$; b, e). In all panels, * and ** represent $P < 0.05$ and $0.01$, respectively, with two-way ANOVA and following the Dunnett's test for multiple comparison analysis ($n = 3$–$5$ in dots; with reference to Ang II or Parent). **g–i**, Schematic representation (g), microscopic imaging (h), and quantifications (i) of the NanoBiT-BRET assay to assess proximity of the three AT1R-Gq-GRK components. Gαq-Lg, Sm-GRK6 and AT1R-mVenus were expressed in the parent cells. The complemented NanoBiT luciferase signal and the BRET signal via mVenus were imaged before and after the indicated stimulation conditions (h). Fold-changes of NanoBiT, mVenus and BRET ratio index were indicated in (i). * and ** represent $P < 0.05$ and $0.01$, respectively, with the Tukey's test among 4 groups ($n = 17$–$21$ cells). In Fig. 6b, e, lines and shaded regions represent mean and SEM, respectively, of 3–5 independent experiments with each performed in duplicate. In Fig. 6c, f, Bars and error bars represent mean and SEM, respectively, of 3–5 independent experiments with each performed in duplicate. In Fig. 6h, representative images of 17–21 cells were shown. In Fig. 6i, shaded regions represent a histogram of 20–29 cells. In Fig. 6c, f, i, * and ** represent $P < 0.05$ and $0.01$, respectively, with two-way ANOVA followed by the Dunnett's test (c, f) or the Tukey HSD test (i) for multiple comparison analysis (c, f; with reference to the Ang II stimulation). See Supplementary statistics data file for additional statistics and exact $P$ values.

NanoBiT and bioluminescence resonance energy transfer (BRET) strategy[47,48]. We used GRK6 because of its higher basal complementation signal than GRK5 (Supplementary Fig. 17a). Prior to ligand stimulation, assembly of the three components was detected as a BRET signal whereas this signal was undetectable in the control conditions (Fig. 6g and Supplementary Fig. 17f–j). Ang II stimulation attenuated both NanoBiT-derived intensity and mVenus-mediated BRET intensity, whereas the BRET ratio was slightly increased (Supplementary Movies 3). The TRV/Ang II + YM treatments showed modest effects in both NanoBiT signal and BRET signal, thus leaving the BRET ratio unchanged (Fig. 6h, i). These results indicate that the three components are in close proximity to each other in the basal state, likely in the immobile domain as well (Fig. 7), and that, upon Ang II stimulation, Gq and GRK6 individually and simultaneously, rather than as the Gq-GRK6 complex, dissociates from AT1R.

Finally, we examined whether the Gq-regulated GRK-subtype selectivity is generalizable to other Gq-coupled receptors. We found that both of the tested Gq-coupled receptors (the serotonin 5-HT$_{2A}$ receptor and the neurotensin NTS$_1$ receptor) showed a gain of GRK5/6-mediated β-arrestin recruitment upon YM treatment, whereas the effect was completely absent in the tested non-Gq-coupled receptors (the Gs-coupled β2 adrenergic receptor and the Gi-coupled μOR opioid receptor) (Supplementary Fig. 18).

## Discussion

The bidirectional regulatory relationship between G proteins and β-arrestin has received considerable attention in prior studies. The classical role of β-arrestin is to suppress G-protein coupling by competitively engaging in the receptor core[49]. As a signal enhancer, β-arrestin induces sustained G-protein signaling by forming "megaplex," a ternary GPCR-G-protein-β-arrestin complex[50]. Oppositely, G-protein activation triggers the initiation of the β-arrestin system by releasing free Gβγ subunits and activating GRK2/3[14]. A GTP-bound, active Gαq subunit is shown to bind to GRK2/3[17], yet its functional effect has been only observed in the Gαq side (enhancement of GTPase activity)[18]. To the best of our knowledge, no specific effect on GRKs is reported for the Gα subunits nor is negative feedback regulation of β-arrestin by G proteins. In this study, we demonstrate that the Gq heterotrimer acts as a negative modulator of β-arrestin by suppressing the function of GRK5/6, in turn enhancing the access of GRK2/3 to AT1R, which collectively affects phosphorylation code-sensitive β-arrestin function.

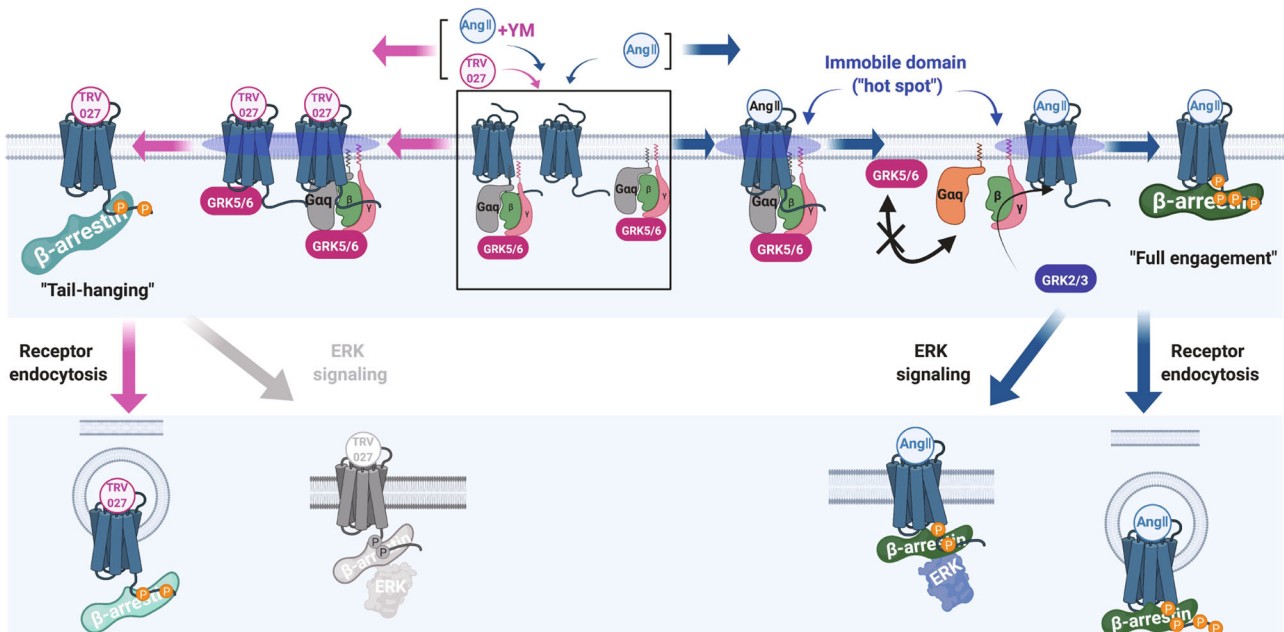

**Fig. 7 Molecular mechanism of Gq-GRK5/6 interplay, regulation by biased ligands and effects on β-arrestin functions.** Upon ligand binding, following events occur depending on Gq activation state. 1. Gq and GRK5/6 form a complex in the Gq-inactive state (box in the middle). 2. Upon TRV/Ang II + YM treatments, the AT1R-Gq-GRK5/6 complex is accumulated in the immobile hot spot, where AT1R undergoes phosphorylation by GRK5/6 (left). 3. Upon Ang II stimulation, the AT1R-Gq-GRK5/6 complex is transient and disassembled through the Gq activation, probably in the hot spot (right). 4. The active Gαq and GRK5/6 become unbound from each other and are repelled from the hot spot (right). 5. GRK2/3 can engage with AT1R, especially in the presence of free Gβγ produced from activated Gq (right). 6. GRK5/6-phosphorylated AT1R induces tail-hanging β-arrestin conformation (left), causing receptor endocytosis, but not ERK signaling (bottom left). 7. Differentially phosphorylated AT1R, mainly by GRK2/3 and additionally by GRK5/6, induces fully engaged β-arrestin conformation (right), allowing both receptor endocytosis and ERK signaling (bottom right).

Our results reveal a novel mechanism underlying GRK-subtype selectivity and subsequent transducer bias at β-arrestin (Fig. 7). Previous dual-color SMT analysis for GPCRs and G proteins[22,46] along with single-particle tracking by photo-activation localization microscopy (sptPALM) analysis of β-arrestin[51] demonstrate that, upon ligand stimulation, active GPCRs and G proteins accumulate in immobile domains or "hot spots." Here, we report, for the first time, dual-color SMT analysis for a GPCR and GRKs and observe that the AT1R and GRK2/GRK5 molecules are concentrated in similar domains. We propose that this phenomenon is the molecular basis for efficient receptor phosphorylation. Furthermore, we speculate that the high density of AT1R, Gq and GRK5/6 in the immobile domain allows a complex formation of these molecules and, once Gq is activated, it triggers separation of the complex and dispersal of Gq and GRK5/6 from the domain. Additionally, active Gq facilitates GRK5/6 translocation from the plasma membrane to the cytosol by inducing $Ca^{2+}$-mediated calmodulin activation[15] and in turn recruits and activates GRK2/3 by releasing free Gβγ[14]. Overall, Gq activity acts as a molecular switch in GRK-subtype selectivity.

The functional diversity of individual GRKs has been well-studied for the β2AR. In cultured cells, isoproterenol, a reference ligand for β2AR, induces phosphorylation of β2AR by all of the major GRK subtypes, i.e., GRK2, GRK3, GRK5, and GRK6[11]. On the other hand, carvedilol, a unique ligand for β2AR, favors GRK5 and GRK6 for β2AR phosphorylation[11]. A previous study on phospho-site mapping showed that GRK2 and GRK6 phosphorylate the distal and the proximal sites, respectively, in the C-terminal tail of β2AR, demonstrating that isoproterenol and carvedilol provide different phosphorylation codes. In accordance with the β2AR study, our Ala mutant analysis indicated that Ser326, the most proximal Ser/Thr site in AT1R, is selectively phosphorylated by GRK5/6 (Supplementary Fig. 3k, l). In mice,

carvedilol improves cardiac function via β-arrestin2[52]. Isoproterenol, on the other hand, worsens cardiac dysfunction in a β-arrestin2-dependent manner[53]. These reports indicate that pharmacologically distinct ligands elicit different β-arrestin-dependent responses by preferentially utilizing specific GRKs. However, it remains unclear exactly how, at a molecular level, GRKs influence β-arrestin conformations and functions. Using AT1R as a model, our study is the first to provide clear evidence showing that two ligands, by utilizing different GRKs, induce distinct engagement of β-arrestin, leading to different AT1R-mediated cellular responses.

Pharmacological and genetic studies in mice have demonstrated distinct roles of GRKs in AT1R-involved myocardial contraction. Administration of Ang II or SII, a TRV027-like ligand, causes myocardial contraction via β-arrestin2[54]. Intriguingly, Ang II-induced and SII-induced phenomena were abolished in GRK2-deficient mice and GRK6-deficient mice, respectively, indicating that the biased ligand has GRK preference[54]. Our cell-based study also demonstrates that SII along with TRV027 show preference toward GRK5/6 while Ang II preferentially utilizes GRK2/3. Furthermore, although our study shows that β-arrestin-mediated pERK is different between Ang II and TRV027, among a diverse set of β-arrestin effector molecules[55], there may be another β-arrestin function that is shared between the two ligands and mediates myocardial contraction.

It has previously been assumed that different structural states of β-arrestin induce distinct functions. Structural studies using negative-stain EM[19] and recent cryo-EM[20,56,57] have shown that β-arrestin has at least two distinct modes with GPCR interactions: the tail hanging (engagement in a phosphorylated receptor via its C-terminus) and the core engagement (C-terminal interaction plus deep binding to the receptor core via the finger-loop region

of β-arrestin; here termed "full engagement"). The tail-hanging mode and the full engagement mode are called partially active and fully active, respectively, based on a proposed stepwise activation mechanism of β-arrestin[19]. A discrete mechanism, called catalytic activation, is documented for the β1 adrenergic receptor (β1AR)-induced β-arrestin activation by a study using total internal reflection fluorescence (TIRF) microscopy[51]. In this model, β-arrestin binding to β1AR is a transient event, but its active state sustains even after dissociation from the receptor. Unlike the tail-hanging and full engagement modes, this interaction does not rely on phosphorylation of the β1AR C-tail, but instead requires binding to the β1AR core (here termed "tail-free core engagement"). Thus, at least three modes of a GPCR-β-arrestin complex are present.

Although the GPCR-β-arrestin-binding modes have been documented in significant detail as described above, their functional distinctions remain less clear. The tail-hanging and the full engagement modes of β-arrestin are capable of inducing both receptor internalization and β-arrestin-dependent Src signaling[21]. The tail-free core-engagement mode of β-arrestin triggers ERK activation[51]. However, no clear functional difference has previously been reported, in part owing to a lack of knowledge about biased ligands that induce different β-arrestin-engagement modes.

In this study, we described different engagement modes of β-arrestin2 by the three AT1R-stimulated conditions. Ang II-induced all three modes, whereas the TRV/Ang II + YM treatments caused the occurrence of the two binding modes, except for the core-engagement mode (Supplementary Figs. 3c, d, 9a–i). These results support the notion that the phosphorylation state of AT1R is important in the transition from the tail-hanging mode to the full engagement mode. The lack of full engagement in the Ang II + YM condition suggests that receptor phosphorylation by GRK5/6 alone is insufficient for binding of β-arrestin to the receptor core and that additional phosphorylation by GRK2/3 is required. Together, our sensor-based analysis provides an insight into how the distinct β-arrestin-engagement modes are regulated.

Importantly, we observed functional differences among these modes. First, we showed that receptor phosphorylation is necessary and sufficient for AT1R internalization. Interestingly, Ang II stimulation caused endosomal translocation of β-arrestin even under the GRK-depleted condition (Supplementary Fig. 6f–k), whereas AT1R itself remained on the cell surface (Fig. 3a–c and Supplementary Fig. 6a–e). This suggests that AT1R internalization requires binding of β-arrestin to the phosphorylated receptor tail and that active β-arrestin, via the catalytic activation mechanism, is translocated to the endosome without forming a stable AT1R-β-arrestin complex. Second, we observed differences in β-arrestin-dependent signaling (Fig. 3d). Among the three conditions, Ang II potently induced pERK response, highlighting a signaling role of the full-engaged β-arrestin from the others. Together, our results elucidate the relationship of the β-arrestin-engagement modes with their functional outcomes.

Finally, we observed functional similarities and differences between β-arrestin1 and β-arrestin2. Although numerous signaling molecules have been reported to bind to both β-arrestin1 and β-arrestin2, it is generally assumed that β-arrestin1 plays a role in desensitization and internalization whereas β-arrestin2 does so in signaling. In this regard, many physio-pharmacological effects are attributed specifically to β-arrestin2[58]. In this study, we found that both β-arrestin subtypes showed virtually indistinguishable functionalities including GRK subtype preference, receptor endocytosis and ERK phosphorylation with the exception of β-arrestin2 being capable of the tail-free core engagement (Supplementary Fig. 3c, d, Supplementary Fig. 9a–c). These observations suggest that β-arrestin1 and β-arrestin2 mostly function redundantly and that GPCRs that are not susceptible to GRK-mediated phosphorylation or that lack the phosphorylation code[59,60], depend on β-arrestin2 due to their inability to associate with β-arrestin1. Altogether, this study clearly links the mechanisms for activation of multiple β-arrestins and their different functions.

In conclusion, we report a novel regulatory mechanism for GRKs and β-arrestins. In contrast to the previous assumption that G protein and β-arrestin are separable, our finding highlights the close relationship between the two GPCR effectors specifically, through a Gq-regulated GRK switch that leads to transducer bias at β-arrestin. These findings provide a conceptual foundation for biased signaling based drug discovery.

## Methods

**Reagents and plasmids**. Ang II, SI, SII, SVA, and Ang (1–7) were purchased from Peptide Institute. TRV027 was custome-synthesized by Genscript. Gq signaling inhibitors were purchased from FujiFilm Wako Pure Chemical (YM-254890), Cayman (U-73122), Adooq (Go6983) and Sigma-Aldrich (BAPTA-AM). EGF was purchased from Cayman. Isoproterenol was purchased from Sigma-Aldrich.

Plasmids for the NanoBiT-β-arrestin-recruitment assay[61], the NanoBiT-Ib30 β-arrestin conformation assay[44] and the NanoBiT-G-protein-dissociation assay[62] were previously described.

For the HiBiT-based receptor internalization assay, human full-length AT1R was N-terminally fused to a HiBiT cassette (HiBiT-AT1R), which contained the Interleukin 6-derived signal sequence followed by the HiBiT sequence and a linker (MNSFSTSAFGPVAFSLGLLLVLPAAFPAPVSGWRLFKKISGGSGGGGSG; HiBiT tag underlined; gene synthesized with codon optimization). For the NanoBiT-β-arrestin-translocation assay, human full-length β-arrestin1 and β-arrestin2 were N-terminally fused to a small fragment (SmBiT) of the NanoBiT luciferase with a 15-amino acid flexible linker (GGSGGGGSGGSSSGG) and the resulting constructs were referred to as Sm-β-arrestin1 and Sm-β-arrestin2, respectively. For the Ib30 assay, the β-arrestin1 conformation-sensitive intrabody 30[44] was N-terminally fused to SmBiT with the flexible linker and the resulting construct was referred to as Sm-Ib30. A FYVE domain of human Endofin (amino acid regions Gln739-Lys806; gene-synthesized with codon optimization by Genscript)[63] was C-terminally fused to a large fragment (LgBiT) with the flexible linker and the resulting construct was referred to as Endo-Lg.

For the NanoBiT-GRK-recruitment assay, human full-length GRK2 (codon-optimized by Genscript), GRK3 (codon-optimized by Genscript), GRK5 (native) and GRK6 (native) were C-terminally fused to LgBiT with the flexible linker (referred to as GRK2-Lg, GRK3-Lg, GRK5-Lg and GRK6-Lg, respectively) and the AT1R was C-terminally fused to the SmBiT with the flexible linker (AT1R-Sm). For the NanoBiT-Gq-GRK proximity assay, LgBiT was inserted into the helical domain (between the residue 123–124; within the loop connecting the αB and the αC helices) of the human Gαq subunit flanked by the flexible linkers (Gαq-Lg). GRK5 and GRK6 were N-terminally fused to SmBiT with the flexible linker (Sm-GRK5 and Sm-GRK6, respectively). For the NanoBiT-BRET imaging, AT1R was C-terminally fused the SmBiT to mVenus with the flexible linker (AT1R-mVenus).

For the flow cytometry analysis, human full-length AT1R were N-terminally fused to the FLAG epitope tag with the preceding hemagglutinin-derived signal sequence (MKTIIALSYIFCLVFADYKDDDDKGGSGGGGSGGSSSGGG; FLAG epitope tag underlined). The FLR truncation (corresponding to the amino acid regions Gly64-Thr74 and Gly65-Thr75 in β-arrestin1 and β-arrestin2, respectively) was introduced according to a previous study[21]. For Δtail-phos AT1R mutant, all twelve Ser and Thr in the C-terminal region of the AT1R (Supplementary Fig. 9e, f) were replaced with Ala (amino acid regions Pro321-Glu359; PPKAKAHANLAAKMAALAYRPADNVAAAAKKPAPCFEVE; Ala substitutions underlined). For the phospho-code AT1R mutant, the seven Ser and Thr within the potential phosphorylation codes[59] in the C-terminal region (Supplementary Fig. 5e) were individually replaced with Ala (PPKAKSHSNLSTKMSTLSYRPSDNVSSSTKKPAPCFEVE; Ala-substituted residues are underlined).

Unless otherwise indicated, all of the constructs were inserted into the pCAGGS or the pcDNA3.1 expression plasmid vector.

**Cell culture and transfection**. HEK293A cells (Thermo Fisher Scientific) and their derivative G-protein-deficient HEK293A cells, including ΔGs (*GNAS/GNAL*-deficient), ΔGi (*GNAI1/GNAI2/GNAI3/GNAO1/GNAT1/GNAT2/GNAZ*-deficient), ΔGq (*GNAQ/GNA11*-deficient), ΔG12 (*GNA12/GNA13*-deficient) and ΔARRB (*ARRB1/ARRB2*-deficient)[64–68] were maintained in Dulbecco's Modified Eagle Medium (DMEM, Nissui Pharmaceutical) supplemented with 10% fetal bovine serum (GIBCO, Thermo Fisher Scientific) and penicillin-streptomycin-glutamine (complete DMEM). Transfection was performed by using a lipofection reagent, polyethylenimine (PEI) solution (Polyethylenimine "Max", Polysciences). Typically, HEK293A cells were seeded in a 6-well culture plate at cell density of 2–3 × 10^5 cells per mL in 2 mL (per well hereafter) of the complete DMEM and cultured for one day in a humidified 37 °C incubator with 5% CO$_2$. A PEI transfection

solution was mixed by combining plasmid a solution diluted in 100 μL of Opti-MEM and 5 μL of 1 mg per mL PEI solution in 100 μL of Opti-MEM. The transfected cells were further incubated for one day before subjected to an assay as described below.

**Generation of GRK-deficient cells and GRK/Gq-deficient cells.** GRK-deficient HEK293A cells were generated by mutating genes encoding members of the GRK family from parent HEK293A cells (Thermo Fisher Scientific) using the CRISPR/Cas9 system[65]. sgRNA constructs targeting the *GRK2*, *GRK3*, *GRK5* and *GRK6* genes were designed using a CRISPR design tool (http://crispr.mit.edu) so that a SpCas9-mediated DNA cleavage site (3-bp upstream of the protospacer adjacent motif [PAM] sequence [NGG]) encompasses a restriction enzyme-recognizing site. Designed sgRNA-targeting sequences including the SpCas9 PAM sequences were as follows: 5'-CGAGGTCTATGGGTGCCGGAAGG-3' (for the *GRK2* gene; where the Hap II restriction enzyme site is underlined and the PAM sequence is in bold), 5'-TTATTGGACGAGGAGGAATTCGGG-3' (for the *GRK3* gene; Hinf I), 5'-AATGTATGCCTGCAAGCGCTTGG-3' (for the *GRK5* gene; Hha I), and 5'-GTGCTACTCAAGGCCCGGGAAGG-3' (for the *GRK6* gene; Sma I). The designed sgRNA-targeting sequences were inserted into the BbsI site of the pSpCas9 (BB)-2A-GFP (PX458) vector (a gift from Feng Zhang at the Broad Institute; Addgene plasmid No. 48138). Correctly inserted sgRNA-encoding sequences were verified with a Sanger sequencing (Fasmac). The *GRK2* and the *GRK3* genes were simultaneously targeted by using a mixture of the sgRNA plasmids. Similarly, the *GRK5* and the *GRK6* genes were simultaneously targeted. To generate quadruple GRK2/3/5/6-deficient cells, we performed a two-step CRISPR/Cas9-mediated mutagenesis with the first round mutating the *GRK5* and the *GRK6* genes and the second round mutating the GRK2 and the GRK3 genes from the validated GRK5/6-deficient clone (ΔGRK5/6 CL1). Briefly, HEK293A cells were seeded into a 10-cm culture dish and incubated for 24 h before transfection. The PX458 plasmid encoding the sgRNA and SpCas9-2A-GFP was transfected into the cells using Lipofectamine 2000 (Thermo Fisher Scientific). Three days later, cells were harvested and processed for isolation of GFP-positive cells (~5% of cells) using a fluorescence-activated cell sorter (SH800; Sony). After expansion of clonal cell colonies with a limiting dilution method, clones were analyzed for mutations in the targeted genes by restriction enzyme digestion. Candidate clones that harbored restriction enzyme-resistant PCR fragments were further assessed for their genomic DNA alterations by direct sequencing or TA cloning[65]. PCR primers to amplify the sgRNA-targeting sites were as follows: 5'-GTAAATATGTGGCAAGGATGGC-3' and 5'-TCCCCGAGGTATCCCACC-3' (*GRK2*); 5'-TTGTGTTTGGATTTCCCAGTTGAC-3' and 5'-GCCTA-CAGCTTATTTCTTTTGGAGG-3' (*GRK3*); 5'-TCTGACCCCATCCATTCTC-TAC-3' and 5'-GATGCTCACTCACCACAAACTG-3' (*GRK5*); 5'-GAGAACATCGTAGCGAACACG-3' and 5'-AGGTGCGGAGGAGGAAGAC-3' (*GRK6*).

HEK293 cells deficient for the Gq family members and the GRK family members (GRK2, GRK3, GRK5 and GRK6) were generated by mutating the GNAQ and the GNA11 genes (encoding Gαq and Gα11, respectively) from the ΔGRK2/3/5/6 (CL2) cells using previously validated sgRNA sequences for the two genes[66]. We transfected a mixture of *GNAQ* sgRNA-containing pX330 (a gift from Feng Zhang at the Broad Institute; plasmid No. 42230) and *GNA11* sgRNA-containing PX458 plasmids (sgRNA-targeting sites and the SpCas9 PAM sequences are 5'-AAACAAGAAAGATCTTCTAGAGG-3' for GNAQ [Xba I] and 5'-AGGGTACTCGATGATGCCGGTGG-3' for GNA11 [Hap II]). By following the procedure as described above, we isolated clonal cells and screened for their genotype by the restriction-enzyme method. Successful targeting was confirmed by the immunoblot analysis.

**Quantitative real-time PCR.** Total RNA of HEK293 cells was isolated using a GenElute Mammalian Total RNA Miniprep Kit (Sigma-Aldrich). The purified RNA was reverse-transcribed using a High-Capacity cDNA Reverse Transcription Kit (Thermo Fisher Scientific) according to manufacturer instructions. Real-time quantitative PCRs were performed with SYBR Premix Ex Taq (Takara Bio) and monitored by ABI Prism 7300 (Applied Biosystems). ΔΔCT method was used to calculate relative expression levels of cDNA samples. Primers were as follows: *GRK1*, 5'-CACAGCAGGTTCATCGTGTC-3' and 5'-AGCCAGGGTTCTCCTCATTC-3'; *GRK2*, 5'-CAGAAGTACCTGGAGGACCG-3' and 5'-TGGTTCAGGCAGAAGTCTCG-3'; *GRK3*, 5'-CAAGTTCACTAGATTTTGTCAGTGG-3' and 5'-TTCCCCGAATCCTCCTCGTC-3'; *GRK4*, 5'-AGAAGAGGAAAGGTGAAGCTATGG-3' and 5'-GCACCAAGCACAAGGCATC-3'; *GRK5*, 5'-CAATCGGGAGGCTGCTTTTC-3' and 5'-TCCTTCCCTTTCTCTCCCAG-3'; *GRK6*, 5'-GATCTGCTGTGGCCTGGAG-3' and 5'-ACCGTGGTCATCCAGCAAG-3'; GRK7, 5'-GCTGGGGAAAGGTGGTTTTG-3' and 5'-GCCGCTTCTTGTCCAGTTTC-3'; *GAPDH*, 5'-GCCAAGGTCATCCATGACAACT-3' and 5'-GAGGGGCCATCCACAG-3'.

**Western blot.** Parent and GRK-deficient cells were lysed by SDS-PAGE sample buffer (62.5 mM Tris-HCl (pH 6.8), 50 mM dithiothreitol, 2% SDS, 10% glycerol and 4 M urea) containing 1 mM EDTA, 1 mM phenylmethylsulfonyl fluoride and 2 mM sodium orthovanadate. Lysates derived from an equal number of cells were

separated by 8% or 12.5% SDS-polyacrylamide gel electrophoresis. Subsequently, the proteins were transferred to nitrocellulose membrane (GE Healthcare). The blotted membrane was blocked with 5% skim milk-containing blotting buffer (10 mM Tris-HCl (pH 7.4), 190 mM NaCl and 0.05% Tween 20), immunoblot with primary and secondary antibodies. Primary antibodies used in this study were anti-GRK2 rabbit monoclonal antibody (R&D, MAB43391, 2089B, lot CKLB021901A, 1:2000 dilution), anti-GRK3 rabbit monoclonal antibody (CST, #80362, D8G6V, lot 80362 S, 1:1000 dilution), anti-GRK5 mouse monoclonal antibody (Santa Cruz Biotechnologies, sc-518005, D-9, lot J3117, 1: 5000 dilution), anti-GRK6 rabbit monoclonal antibody (CST, #5878, D1A4, lot 5878 S, 1:1000 dilution), anti-α-tubulin mouse monoclonal antibody (Santa Cruz Biotechnologies, sc-32293, DM1A, lot K1414, 1:2000 dilution), anti-Flag-epitope tag mouse monoclonal antibody (Wako Pure Chemicals, 014-22383, Clone 1E6, 1:1000 dilution), anti-Gs/olf mouse monoclonal antibody (Santa Cruz Biotechnologies, sc-55546, E-7, lot B2307, 1:1000 dilution), anti-Gi mouse monoclonal antibody (Neweast biosciences, 26003, lot. PH063, unknown clone; http://www.neweastbio.com/AntiGTPase/23, 1:1000 dilution), anti-Gq goat polyclonal antibody (abcam, ab128060, D-6, lot GR108939-6, 1:2000 dilution), anti-G11 mouse monoclonal antibody (Santa Cruz Biotechnologies, sc-390382, D-6, lot D0113, 1:2000 dilution), anti-G13 rabbit monoclonal antibody (abcam, ab128900, EPR5436, GR90067-7, 1:1000 dilution), anti-Gbeta mouse monoclonal antibody (Santa Cruz Biotechnologies, sc-166123, H-1, lot F2718, 1:1000 dilution), anti-Ggamma mouse monoclonal antibody (Santa Cruz Biotechnologies, sc-166419, C-5, lot E1517, 1:1000 dilution), anti-beta arrestin1 rabbit monoclonal antibody (CST, #12697, D8O3J, lot 1, 1:1000 dilution), anti-beta arrestin2 rabbit monoclonal antibody (CST, #3857, C16D9, lot 2, 1:1000 dilution), anti-Phospho-p44/42 MAPK(T202/Y204) rabbit monoclonal antibody (CST, #8544, D13.14.4E, lot 3, 1:1000 dilution) and anti-p44/42 MAPK rabbit monoclonal antibody (CST, #4695, 137F5, lot 2, 1:10000 dilution). Secondary antibodies that were conjugated with horseradish peroxidase (HRP) were anti-mouse IgG, HRP-Linked F(ab')2 Fragment Sheep (GE Healthcare, NA9310, lot F17365253), anti-Rabbit IgG, HRP-Linked F(ab')2 Fragment Donkey (GE Healthcare, NA9340, lot 17041889) and anti-goat IgG (American Qualex, A201PS lot 7A0327H). Membrane was soaked with a luminol reagent (100 mM Tris-HCl (pH 8.5), 50 mg per mL Luminol Sodium Salt HG (FujiFilm Wako Pure Chemical), 0.2 mM *p*-Coumaric acid and 0.03% (v/v) of $H_2O_2$). A chemiluminescence image was acquired and band intensity was analyzed with Amersham Imager 680 (Cytiva).

**NanoBiT assays.** β-arrestin recruitment to AT1R was measured by the NanoBiT β-arrestin-recruitment assay[62]. The parent and the GRK-deficient cell lines were transfected with a mixture of 100 ng Lg-β-arrestin and 500 ng AT1R-Sm plasmids (per well in a 6-well plate unless otherwise noted). After incubation for 1 day, transfected cells were collected with 0.5 mM EDTA-containing Dulbecco's PBS (D-PBS), centrifuged and suspended in 2 mL of HBSS containing 0.01% bovine serum albumin (BSA; fatty-acid-free grade; SERVA) and 5 mM HEPES (pH 7.4) (assay buffer). The cell suspension was dispensed in a white 96-well plate at a volume of 80 μl per well (hereafter 96-well plate) and loaded with 20 μl of 50 μM coelenterazine (CTZ, Carbosynth) diluted in the assay buffer. After 2-h incubation at room temperature, the plate was measured for baseline luminescence (SpectraMax L equipped with 2PMT, Molecular Devices; SoftMax Pro Version 7.03, Molecular Devices) and 20 μl of 6× ligand diluted in the assay buffer or the assay buffer alone (vehicle) were manually added. The plate was read for 15 min with an interval of 20 sec or 40 sec at room temperature. The luminescence counts from 13 min to 15 min (unless otherwise indicated) after ligand addition were averaged and normalized to the initial counts. We note that due to the time lag between the manual ligand addition and start of plate reader measurement (typically, 30–60 s), there were already changes for some of the early responding sensors (e.g., GRK recruitment and GRK-Gβγ association) at the first measurement (labeled as 0 min in the x-axis).

Gq activation was measured by the NanoBiT-G-protein-dissociation assay[62] with minor modifications. Plasmid transfection was performed by using a mixture of 100 ng Gαq with LgBiT insertion at the loop between the αA and the αB helices (the residue 97–98), 500 ng Gβ1, 500 ng SmBiT-fused Gγ2 (C68S), 100 ng RIC8A and 200 ng AT1R plasmids (per well in a 6-well plate). The transfected cells were subjected to the NanoBiT assay as described in above and the luminescence counts from 3 min to 5 min after ligand addition were used for quantification.

An active β-arrestin conformation was detected by the NanoBiT-Ib30 assay[44] with minor modifications. Transfection was performed by using a mixture of 500 ng Sm-Ib30, 500 ng Lg-β-arrestin and 200 ng GPCR plasmids. The transfected cells were subjected to the NanoBiT assay as described above.

β-arrestin endosomal translocation was measured by a bystander approach[63], but with a NanoBiT modification. Transfection was performed by using a mixture of 500 ng Endo-Lg, 100 ng Sm-β-arrestin and 200 ng GPCR plasmids. The transfected cells were subjected to the NanoBiT assay as described in above.

GRK recruitment to AT1R was measured by using the Lg-GRK and the AT1R-Sm constructs. Transfection was performed by using a mixture of 200 ng Lg-GRK and 200 ng AT1R-Sm plasmids. The transfected cells were subjected to the NanoBiT assay as described in above and the luminescence counts from 3 to 5 min after ligand addition were used for quantification.

For the Gαq-GRK proximity assay, transfection was performed by using a mixture of 500 ng Gαq-Lg, 500 ng Sm-GRK5 or Sm-GRK6, 500 ng Gβ1, 500 ng Gγ2, 100 ng RIC8A and 200 ng AT1R plasmids. For the Gβγ-GRK proximity assay, transfection was performed by using a mixture of 100 ng GRK5-Lg or GRK6-Lg, 500 ng Sm-Gβ1, 500 ng Gγ2, 100 ng RIC8A and 200 ng AT1R plasmids. The transfected cells were subjected to the NanoBiT assay as described in above and the luminescence counts from 3 to 5 min after ligand addition were used for quantification.

For downstream of Gq activation, we used the NanoBiT-IP$_3$ assay[57]. Transfection was performed by using a mixture of 1 µg Lg-IP3R2-Sm and 200 ng AT1R plasmids. For downstream of G12 activation, we used the NanoBiT-RhoA assay[57]. Transfection was performed by using a mixture of 100 ng Lg-RhoA, 500 ng Sm-PKN1-GBD and 200 ng AT1R plasmids. The transfected cells were subjected to the NanoBiT assay as described in above and the luminescence counts from 3 min to 5 min after ligand addition were used for quantification.

For the GRK addback experiments using ΔGRK2/3/5/6 cells and the GRK/Gq-6KO cells, 1 ng GRK2, 10 ng GRK3, 5 ng GRK5 and 5 ng GRK6 (per well in a 6-well plate; untagged or C-terminally FLAG-tagged constructs) were co-expressed along with indicated constructs.

**HiBiT-based receptor internalization assay**. AT1R internalization was measured by HiBiT-based receptor internalization assay[69]. The parent and the GRK-deficient cell lines were transfected with 100 ng HiBiT-AT1R plasmid (per well in a 6-well plate). For the experiments using the β-arrestin1/2-deficient cells, ΔARRB cells were transfected with 100 ng HiBiT-AT1R plasmid and 100 ng Sm-β-arrestin plasmid. After incubation for 1 day, cells were harvested, suspended in 1 mL of the assay buffer, dispensed in a white 96-well half-area plate at a volume of 25 µL per well, and mixed with 25 µL of 2× substrate buffer consisting of 1:200 of a LgBiT stock solution (Promega) and 20 µM furimazine in the assay buffer. After 40 min incubation at room temperature, the plate was measured for baseline luminescence, and a titrated ligand (10 µL) diluted in the 1× substrate buffer was manually added. The plate was immediately read at room temperature for the following 30 min at a measurement interval of 30 sec. The luminescence counts from 27 min to 30 min after ligand addition were averaged and normalized to the vehicle stimulation count.

**AlphaLISA-based ERK1/2 phosphorylation assay**. ERK1/2 activation was monitored with the AlphaLISA assay (PerkinElmer). The parent and ΔARRB cells were transfected with 200 ng AT1R plasmid (per well in a 6-well plate) with or without 100 ng Sm-β-arrestin plasmid. After incubation for 1 day, transfected cells were collected, suspended in the complete DMEM at a concentration of $10^5$ cells per ml and reseeded into a 96-well plate ($10^4$ cells per well with 100 µL). One day later, the cells were serum-starved in DMEM overnight. The cells were then stimulated for 30 min with ligands and lysed in 100 µL of a lysis buffer. The lysate was diluted 1:10 and analyzed according to a manufacturer's protocol. Alpha counts were read on the EnSpire microplate reader (PerkinElmer).

**Glosensor cAMP assay**. Plasmid transfection was performed in a 6-well plate with a mixture of 1 µg of Glo-22F cAMP biosensor (human codon-optimized and gene-synthesized)-encoding pCAGGS plasmid and 200 ng of the human vasopressin V2R-encoding plasmid or the human opioid µOR plasmid. After 1-day incubation, the transfected cells were harvested with 0.53 mM EDTA-containing D-PBS, centrifuged at 190 g for 5 min and suspended in the assay buffer described in the NanoBiT assay (vehicle; 1 mL per well). The cells were seeded in a half-area white 96-well plate (30 µL per well) and loaded with D-luciferin potassium solution (10 µL of 8 mM solution per well; FujiFilm Wako Pure Chemical). After 2 h incubation in the dark at room temperature, the plate was read for its initial luminescent count (SpectraMax L, Molecular Devices). For the Gs assay, the V2R-expressing cells were treated with vehicle, 100 nM arginine vasopressin (AVP, Peptide Institutes) or 10 µM forskolin (FujiFilm Wako Pure Chemical) (10 µL of 5X solution per well). For the Gi assay, the µOR-expressing cells were treated with 10 µM forskolin (vehicle) or forskolin-supplemented [D-Ala2, N-Me-Phe4, Gly5-ol]-Enkephalin acetate salt (DAMGO at 1 µM; Sigma-Aldrich). Thereafter, kinetics were measured for 20 min and expressed as fold-change values. Fold-change luminescent signals from 17 min to 20 min after compound addition were averaged and normalized to those in forskolin-treated condition.

**TGF-α shedding assay**. Gq activation was measured by TGF-α shedding assay[70]. Plasmid transfection was performed in a 6-well plate with a mixture of 500 ng of the alkaline phosphatase-tagged-TGF-α (AP-TGF-α) reporter-encoding pCAGGS plasmid and 200 ng of the human histamine H1 receptor. After 1-day culture, the transfected cells were harvested by trypsinization, pelleted by centrifugation at 190 g for 5 min and washed once with the HBSS containing 5 mM HEPES (pH 7.4). After centrifugation, the cells were resuspended in 6 mL of the HEPES-containing HBSS. The cell suspension was seeded in a clear 96-well culture plate (cell plate) at a volume of 90 µl (per well hereafter) and incubated for 30 min in a 5% $CO_2$ incubator at 37 °C. The cells were treated with 10 µM histamine (diluted in HBSS containing 5 mM HEPES (pH 7.4) and 0.01% (w/v) BSA). After spinning the cell plates, conditioned media (80 µL) was transferred to an empty 96-well plate

(conditioned media (CM) plate). AP reaction solution (10 mM p-nitrophenyl-phosphate (p-NPP), 120 mM Tris–HCl (pH 9.5), 40 mM NaCl, and 10 mM MgCl$_2$) was dispensed into the cell plates and the CM plates (80 µL). Absorbance at 405 nm (Abs$_{405}$) of the plates was measured, using a microplate reader (SpectraMax 340 PC384, Molecular Devices), before and after 1-h incubation at room temperature. Ligand-induced AP-TGF-α release signal was calculated by subtracting spontaneous AP-TGF-α release signa from total ligand-induced AP-TGF-α release response.

For the Gq addback experiments using the GRK/Gq-6KO cells, 100 ng of the untagged Gα subunit plasmids was individually expressed along with the AP-TGFα and the H1 receptor as above.

**Flow cytometry**. HEK293A cells were seeded in a 12-well culture plate at a concentration of $2 × 10^5$ cells per mL (1 mL per well) 1 day before transfection. Transfection was performed by using 100 ng of FLAG-AT1R plasmid. After incubation for 1 day, the cells were collected by adding 100 µL of 0.53 mM EDTA-containing D-PBS, followed by 100 µL of 5 mM HEPES (pH 7.4)-containing HBSS. The cell suspension was transferred in a 96-well V-bottom plate and fluorescently labelled by using anti-Flag-epitope tag monoclonal antibody (Clone 1E6, FujiFilm Wako Pure Chemical; 10 µg per mL) diluted in 2% goat serum- and 2 mM EDTA-containing D-PBS (blocking buffer) and a goat antimouse IgG secondary antibody conjugated with Alexa Fluor 488 (Thermo Fisher Scientific; 10 µg per mL in diluted in the blocking buffer). After washing with D-PBS, the cells were resuspended in 200 µL of 2 mM EDTA-containing-D-PBS and filtered through a 40-µm filter. Fluorescent intensity of single cells was quantified by an EC800 flow cytometer equipped with a 488-nm laser (Sony). Live cells were gated with a forward scatter ('FS-Peak-Lin') cutoff of 390 setting a gain value of 1.7. Mean fluorescence intensity from all of the recorded events (approximately 20,000 cells per sample) was analyzed by a FlowJo software (FlowJo) and used for statistical analysis.

**Single-molecule imaging analysis**. Plasmid DNAs (pFC15A vector, 0.05 µg per 6-cm dish) encoding HaloTag-fused AT1R and SNAP-tag-fused GRK were transfected into parent, ΔGq, ΔGRK2/3/5/6 cells by Lipofectamine 3000 on the day before imaging. The HaloTag and SNAP-tag were labeled for 15 min at 37 °C by 30 nM SF650 HaloTag ligand and 100 nM SNAP-Cell TMR-Star ligand (New England Biolabs), respectively. After 5 min wash with 3 mL 10% FBS/DMEM/F12 three times, the cells on a coverslip were mounted on the Attofluor cell chamber. After washing with 400 µL 0.01% BSA-HBSS three times, cells were incubated 15 min RT in 400 µL Vehicle (0.1% DMSO) or 1 µM YM-254890 in 0.01% BSA-HBSS. Then, 100 µL ligand solution (Vehicle, 1 µM Ang II, or 1 µM TRV027 at final concentration) in 0.01% BSA-HBSS was added into the chamber 15 min before imaging.

Single-molecule images were taken by a homemade TIRF microscope system (lasers: Compass 315M-100 and OBIS 637 nm, Coherent, microscope: TiE, Nikon, autofocus system: PAF, Zido, dichroic: ZT532/640rpc, Chroma, objective: PlanApo 60×, NA 1.49, Nikon, 4× relay lens, Nikon, two-channel imaging system: M202J, Nikon, dichroic: FF640-FDi01, Semrock, emission filters: FF01-585/40 and FF01-676/29, Semrock, two cameras: ImagEM, Hamamatsu). The TIRF microscope system was controlled by AIS, Zido (https://eng.zido.co.jp/). We took 300 frame dual-color movies per cell with 30.5 ms exposure time between 15 and 30 min after ligand stimulation (512×512 pix images, pixel size: 67 nm/pix).

In the single-molecule imaging, we visually screened the imaging regions where the density of particles (for each color) was less than ~1 particle per µm$^2$, an upper-threshold that one can visualize single-particles of fluorescently labeled proteins. In the meantime, a too low density of particles prevents accurate quantification. We typically observed regions contained 0.3~0.6 particles per µm$^2$. Once we chose a cell that expressed the optimal particle density, we unbiasedly took 300 frame dual-color movies per cell with 30.5 ms exposure time between 15 and 30 min after ligand (or vehicle) stimulation (512 × 512 pix images, pixel size: 67 nm/pix). We performed two independent microscopy experiments, for each experimental condition, with recording of 10–15 single cells per coverslip.

In our analysis procedure, after quality checking of recorded movies (e.g., those with out of focus or frame dropping during video recording were omitted), no intentional data exclusion was made. The image processing (subtract background with 25 pixels rolling ball radius, and two frames running average with Running_ZProjector plugin, http://valelab.ucsf.edu/~nstuurman/ijplugins/) were performed by ImageJ with batch processing macro (https://github.com/masataka-yanagawa/ImageJ-macro-ImageProcessingSMT). The alignment between the two channels and SMT analysis were performed using AAS, Zido[71]. The VB-HMM analysis was performed by a LabVIEW-based homemade program or AAS[46,71]. The four-state model was selected that showed the highest lower bound in the VB method (Supplementary Fig. 10g). All subsequent analyses (diffusion dynamics, intensity distribution, colocalization, and statistical analysis) were performed using smDynamicsAnalyzer (https://github.com/masataka-yanagawa/IgorPro8-smDynamicsAnalyzer), an Igor Pro 8.0, WaveMetrix (Igor)-based homemade program. Refer to the reference for the detailed instruction and curve fitting functions[71]. Here, the colocalization of two particles is detected if the two particles are within 100 nm in the same frame, which corresponds to 2~3 SDs of the total position accuracy, and if the two particles are in the same diffusion state. The localization precision of each diffusion state was estimated as shown in Supplementary Fig. 10h based on the equation 6 of Mortensen et al.[72].

**Endocytosis imaging**. Plasmid DNAs (2 µg in total per 6-cm dish) encoding FLAG-AT1R and/or SNAP-tag-fused β-arrestin2 in pcDNA3.1 were transfected into the parent and the GRK-deficient cell lines by Lipofectamine 3000 on the day before imaging. SNAP-tag was stained by 100 nM SNAP-Cell TMR-Star ligand and washed as described above. FLAG-tag was labeled by anti-FLAG monoclonal antibody (clone 1E4; 1:100 in 0.01% BSA-HBSS) for 30 min and washed by 0.01% BSA-HBSS on ice. Then, the cells were stained by antimouse IgG-AlexaFluor 488 (1:200 in 0.01% BSA-HBSS) for 30 min and washed with 0.01% BSA-HBSS on ice. Three µM Go6983 with or without 1 µM YM-254890 were added 15 min before ligand stimulation and cells were brought back to room temperature. Fifteen min after ligand stimulation (Vehicle, 1 µM Ang II, or 1 µM TRV027 at final concentration), the cells were fixed by PBS containing 4% paraformaldehyde and 0.2% glutaraldehyde for 30 min at room temperature. Dual-color images (focus: approximately 2 µm above coverslip, exposure time: 30.5 ms, pixel size: 67 nm/pix) were taken by the same microscope system as described in the single-molecule imaging section with oblique illumination settings. The dichroic and emission filters were changed to the following configurations (lasers: Sapphire 488–200, and OBIS 561 nm, Coherent, dichroic: ZT488/561rpc and T560lpxr, Chroma, emission filters in the two-channel imaging system: ET525/50 m, ET605/70 m, Chroma).

**NanoBiT-BRET imaging**. Plasmid DNAs (1 µg AT1R-mVenus, 0.5 µg Gαq-Lg, 0.5 µg Sm-GRK6, 1 µg Gβ1, 1 µg Gγ2, 0.2 µg RIC8A for 6-cm dish) were transfected into the parental cells by Lipofectamine 3000 on the day before imaging. For control experiments, AT1R-mVenus and Gαq-Lg were replaced with HaloTag-fused AT1R and NanoLuc-fused Gαi1, respectively. The cells on a coverslip are mounted on the Attofluor cell chamber and washed 3 times with 400 µL 0.01% BSA-HBSS. The extracellular fluid was replaced by 300 µL 0.01% BSA-HBSS with vehicle (0.1% DMSO) or 1 µM YM-254890 15 min before imaging. Just before imaging, 100 µL Furimazine solution (5 µL substrate in Nano-Glo Live Cell Assay System, Promega was diluted by 100 µL NanoGlo buffer) was added into the chamber on a microscope.

NanoBiT-BRET time-lapse images were taken by a homemade microscope system with the following configurations (microscope: IX83, Olympus, objective: APON 60×, NA 1.45, Olympus, two-channel imaging system: W-view Gemini-2C, Hamamatsu, dichroic: 470DCXR, Chroma, emission filters: ET450/50, Chroma, BA515–550, Olympus, two cameras: ImagEM X2, Hamamatsu). The microscope system was controlled by the multi-dimensional acquisition program in Metamorph, Molecular Devices. We took dual-color time-lapse images (1024 × 512 pix images, pixel size: 270 nm/pix) with the following settings (exposure time: 30 s, EM gain: 1200, time interval: 2 min, total time: 20 min). The 100 µL ligand solution in 0.01% BSA-HBSS (vehicle, 1 µM Ang II, or 1 µM TRV027 at final concentration) was added between timepoints 2 and 3.

The image processing (split channels, merge channels, crop region of interest, and intensity measurement) was performed by ImageJ with batch processing macro (https://github.com/masataka-yanagawa/ImageJ-macro-ImageProcessingSMT). Cell regions were manually cropped, and measured the mean intensity of each channel ($I_{NanoBiT}$, $I_{Venus}$). Calculation of BRET index, data visualization as violin/box plots and statistical analysis were performed in Igor, where BRET index was defined as follows: BRET index = $I_{Venus}/(I_{NanoBiT} + I_{Venus})$. The fold-change was calculated for each cell based on the intensity ratio between before and approximately 14 min after stimulation.

**Statistical analyses**. Statistical analyses were performed using the GraphPad Prism 8 software (GraphPad) or Igor and methods are described in the legends of the figures. Representation of symbols and error bars is described in the legends. Symbols are either mean values of indicated numbers of independent experiments or numbers of measured cells from a single experiment. Unless otherwise noted, error bars and shaded areas denote SEM. Concentration-response curves were fitted to all data by the *Nonlinear Regression: Variable slope (four parameter)* in the Prism 8 tool with a constraint of the *Hill Slope* of absolute value less than 2. For multiple comparison analysis, one-way or two-way ANOVA and following the Dunnett's test or the Tukey's test or the Sidak test, the multiple *t*-test or two-tailed Welch's *t*-test was used.

**Reporting Summary**. Further information on research design is available in the Nature Research Reporting Summary linked to this article.

## Data availability
All data generated or analyzed during this study are included in this article, its Supplementary Information and Source Data file or from the corresponding authors on reasonable request. The Source data of Figs. 1–6 and Supplementary Fig. 2–18 are provided in the separated Source Data file. Source data are provided with this paper.

## Code availability
The SMT analysis method (smDynamicsAnalyzer: https://github.com/masataka-yanagawa/IgorPro8-smDynamicsAnalyzer) and the image processing methods for SMT and NanoBiT-BRET experiments (https://github.com/masataka-yanagawa/ImageJ-macro-ImageProcessingSMT) are available.

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

## Acknowledgements

We thank Arun K Shukla at Indian Institute of Technology Kanpur for critical discussion and manuscript editing. We also thank Mithu Baidya and Hemlata Agnihotri at Indian Institute of Technology for the Ib30 plasmid; Kayo Sato, Shigeko Nakano and Ayumi Inoue and other members of the laboratory at Tohoku University for their assistance of plasmid preparation and cell-based GPCR assays; Toshifumi Inada and Yoshiro Saito at Tohoku University and Carsten Hoffmann at Friedrich Schiller University Jena for helpful discussion; Hans Bräuner-Osborne at University of Copenhagen for helpful discussion on generation of the GRK-deficient cells. This work was supported by Japan Society for the Promotion of Science (JSPS) KAKENHI grants 17K08264 (A.I.), 21H04791 (A.I.), 21H05113 (A.I.), JPJSBP120213501 (A.I.), JPJSBP120218801 (A.I.) and 20K05760 (Ma.Y.); Grant-in-Aid for JSPS Fellows 19J11256 (K.K.), 20J20669 (Y.O.); Moonshot Research and Development Program JPMJMS2023 (A.I.), PRESTO JPMJPR1331 (A.I.) and JPMJPR20EF (Ma.Y.) from Japan Science and Technology Agency (JST); the PRIME JP19gm5910013 (A.I.), the LEAP JP19gm0010004 (A.I. and J.A.) and the BINDS JP20am0101095 (A.I.) from the Japan Agency for Medical Research and Development (AMED); Daiichi Sankyo Foundation of Life Science (A.I.); The Uehara Memorial Foundation (A.I.).

## Author contributions

Conceptualization, KK and AI; Methodology, KK, Ma.Y, MH, MU, YS, AI; Investigation, KK, Ma.Y, SH, Mi.Y, and YO; Writing, KK, Ma.Y, AI with feedback from all of the coauthors; Funding Acquisition, KK, Ma.Y, YO, MH, MU, YS, JA, AI; Supervision, MU, YS, JA, AI.

## Competing interests

The authors declare no competing interests.
