## [Peer Review File · Nature Communications]

The origin of β -arrestin transducer bias: heterotrimeric Gq as a switch for GRK5/6 selectivityEditorial Note: This manuscript has been previously reviewed at another journal that is not operating a transparent peer review scheme. This document only contains reviewer comments and rebuttal letters for versions considered at *Nature Communications*.

REVIEWERS' COMMENTS

Reviewer #1 (Remarks to the Author):

The authors have adequately addressed the concerns raised in the earlier review. In particular, they have addressed potential alternative interpretations of their findings with new data, and supported their interpretation. The manuscript reports a novel mechanism for the generation of functional selectivity in GPCR signaling that should be of broad interest.

Reviewer #2 (Remarks to the Author):

The authors have addressed most of my comments on the single-molecule imaging and data analysis. I encourage them to pay attention to some of the remaining issues, mostly on clarity. After reading more into the other aspects of the manuscript, I'd like to also echo the other reviewers' suggestions on improving discussions the signaling and the mechanistic model.

1. Regarding the oligomerization states of AT1R and the GRKs, the authors have moved the diffusion trajectories to Extended Data and stated that the oligomer sizes are putative (main text, lines 207-210). This is helpful. To make it flow better, I'd suggest that the authors try something like this in this paragraph. '... (Extended Figure 12). Assuming that both AT1R and the GRKs are monomeric in the fast states are monomeric, we inferred that these molecules are dimers or oligomers in the immobile state. Similarly, AT1R exists as a mixture of dimers and monomers in the slow and the medium states, and GRKs a mixture of dimers and tetramers in these states.'

2. Regarding the colocalization analysis, I am not entirely sure I understand the authors' reasoning as to why they choose to relax the conditions (their response states that it is because AT1R and GRK5 mobility shifted in an opposite way upon Ang II treatment). If AT1R and GRK5 dissociate upon Ang II and now are in different mobility states, then they should no longer considered colocalized. In any case, I think the criterion that the two molecules should be in the same diffusion state should be kept in the colocalization analysis. Thus, I encourage that the authors replace the relevant figure panels (Figs 4d and 4e) with the ones they showed in the response letter (Response Letter Figure 12) and update the text (main text and methods in accordance).

3. I agree with the other reviewers that the authors could further improve discussions on the molecular mechanisms of the observed transducer bias. The new Figure 7 is helpful, but some critical details are absent. For example, when does AT1R phosphorylation by GRK5/6 take place? Should the Gq-GRK5/6 complexes remain bound after the phosphorylation happens? If so, when do the Gq-GRK5/6 complexes fall off the receptor? After arrestin recruitment, endocytosis, etc.? I am aware that these are not directly studied in this work. If answers to these are unclear at present, then they could be question marks on the corresponding steps in the diagram, so the readers are informed. The authors may also want to update the GRK2/3 part of this diagram since GRK2/3 recruitment follows G $\beta\gamma$ dissociation from G α_q . In addition, I wondered if the authors have done pERK assays in DGRK cells (they had endocytosis data from these cells). I do not suggest additional experiments, but if they already have any data on that front, they may consider including those to strengthen the signaling aspects of this work.

4. There are a number of places in the main text where I find the statement confusing. Please consider revising these.

For example, in lines 163-166, it reads 'We note that a large portion of TRV027-induced pERK was sensitive to Go6983 (Extended Data Fig. 7b), indicating a residual Gq-PKC-mediated effect by the biased ligand. Whereas Ang II stimulation induced a robust pERK signal, pERK response was marginal by TRV027 stimulation and nearly undetectable by YM pretreatment (Fig. 3d and Extended Data Fig. 7b-d)'. It would be much easier to follow if the two sentences are swapped in order; even better if the new 2nd sentence reads something like this: 'we note that the low level of pERK induced by TRV207 was largely diminished upon Go6983 treatment, indicative of a residual Gq-PKC-mediated signaling component.'

Lines 169-171, it reads 'Compared with Ang II stimulation, the TRV/Ang II + YM treatments showed decreased levels of Ib30-recognizable β -arrestin1. The TRV/Ang II + YM treatments apparently decreased Ib30 reactivity to β -arrestin1 as compared with Ang II stimulation (Extended Data Fig. 5f, g)'. Are these two sentences redundant?

In lines 195-196, the authors stated that 'Taken together, these results indicate that the GRK-subtype selectivity arises not directly from structural states of the ligand-bound receptor, but from activation level of Gq and, further ...'. There is a gap in the logic flow here (or I missed it somewhere else in the text), which is that YM-pretreatment altered arrestin binding modes on the same Ang II-bound AT1R presumably without altering the structural states of the receptor (since it is the same ligand).

In lines 227-230, the authors wrote 'Collectively, these data support a model in which Gq acts as a negative regulator that causes reduction of GRK5 accessibility to the functional domain and thereby results in the lowered association rate between AT1R and GRK5 (Fig. 4h), whereas it simply activates

GRK2 (Fig. 4i).’ By ‘Gq’ I think the authors meant ‘active Gq’? Also, I am not sure where the authors inferred ‘GRK5 accessibility’ to the functional domain (of the receptor presumably)? This sentence could be written as ‘... in which activation of Gq promotes dissociation of GRK5 from the AT1R and, in the meantime, upregulates GRK2/3 kinase activity via the resulting Gβγ and frees up the receptor for phosphorylation by the activated GRK2/3.’

Reviewer #3 (Remarks to the Author):

The authors have adequately addressed most of the issues I noted during the previous review. A couple of follow-up comments are below (numbered as in the authors’ rebuttal).

1)

a. The authors state in the text that “the GRK5/6-dependent β-arrestin-recruitment response is inversely correlated with Gq activity.” However, the key point they show in the main text Fig 3i is the positive correlation for the ΔGRK5/6 cells (i.e., the β-arrestin-recruitment response that is dependent on GRK2/3). Likewise, in the rebuttal the authors state, “In the plots with five ligands, β-arrestin recruitment in ΔGRK5/6 cells is highly correlated with Gq signaling activity ($R^2 > 0.60$), while the correlation with Gi and G12 signaling activity is weak ($R^2 < 0.21$) (Fig. 1j). These results clearly show that there is an inverse correlation between GRK5/6-mediated β-arrestin recruitment and Gq, but such correlation is subtle for Gi and G12 activity.” Since the ΔGRK5/6 data reflects GRK2/3-dependent effects, and the 5 ligand plot of the ΔGRK2/3 (GRK5/6-dependent) data shows only a weak correlation ($r^2 = 0.41/0.19$), how does this support the authors’ conclusion? Please clarify.

b. The authors exclude AngII from the analysis in the main text since it skews the analysis. However, with AngII excluded, it still appears that a single ligand (SVA) is driving the correlation.

c. Please specify the normalizations in the new panel Fig. 1j (e.g., % AngII max).

3)

a. “Normalization to the β-arrestin-recruitment responses . . .” (Ln 171-172): It appears that these data are normalized to the AngII response in each assay, not to the β-arrestin-recruitment response.

b. A clearer statement would be something to the effect of:

“The TRV/Ang II + YM treatments apparently decreased Ib30 reactivity to β-arrestin1 as compared with Ang II stimulation (Extended Data Fig. 5f, g). Normalization of the Ib30 and β-arrestin-recruitment responses showed that Ib30 recruitment is proportional to the β-arrestin1 recruitment levels (Extended Data Fig. 5h). Thus we conclude that the Ib30-recognizable active β-arrestin conformation is similarly induced in the TRV/Ang II + YM treatments.”

c. Please correct the Extended Fig. references here (now Extended Fig 7 rather than Extended Fig 5).

Below are the authors' point-to-point responses (in blue) to the Reviewers' comments (*italicized and written in black*).

Reviewer #1 (Remarks to the Author):

The authors have adequately addressed the concerns raised in the earlier review. In particular, they have addressed potential alternative interpretations of their findings with new data, and supported their interpretation. The manuscript reports a novel mechanism for the generation of functional selectivity in GPCR signaling that should be of broad interest.

We thank the Reviewer #1 for the valuable comments on our work.

Reviewer #2 (Remarks to the Author):

The authors have addressed most of my comments on the single-molecule imaging and data analysis. I encourage them to pay attention to some of the remaining issues, mostly on clarity. After reading more into the other aspects of the manuscript, I'd like to also echo the other reviewers' suggestions on improving discussions the signaling and the mechanistic model.

1. Regarding the oligomerization states of AT1R and the GRKs, the authors have moved the diffusion trajectories to Supplementary and stated that the oligomer sizes are putative (main text, lines 207-210). This is helpful. To make it flow better, I'd suggest that the authors try something like this in this paragraph. '... (Extended Figure 12). Assuming that both AT1R and the GRKs are monomeric in the fast states are monomeric, we inferred that these molecules are dimers or oligomers in the immobile state. Similarly, AT1R exists as a mixture of dimers and monomers in the slow and the medium states, and GRKs a mixture of dimers and tetramers in these states.'

We thank the Reviewer #2 for the suggestion. To improve readability about the change in the putative oligomer size in each diffusion state upon ligand stimulation (Supplementary Fig. 12c), we have added the following explanation in the same paragraph (Line 203-209).

“Assuming that both AT1R and the GRKs are monomeric in the fast states are monomeric, we inferred that these molecules are dimers or oligomers in the immobile state. Similarly, AT1R exists as a mixture of dimers and monomers in the slow and the medium states, and GRKs a mixture of dimers and tetramers in these states. Ang II stimulation statically significantly decreased the putative oligomer size in GRK5 in the immobile state (Supplementary Fig. 12c). The TRV/Ang II + YM treatments did not alter it and an increase of the putative oligomer size in the medium and fast states of GRK5. In contrast, no significant change was observed in GKR2 (Supplementary Fig. 12c).”

2. Regarding the colocalization analysis, I am not entirely sure I understand the authors' reasoning as to why they choose to relax the conditions (their response states that it is because AT1R and GRK5 mobility shifted in an opposite way upon Ang II treatment). If AT1R and GRK5 dissociate upon Ang II and now are in different mobility states, then they should no longer considered colocalized. In any case, I think the criterion that the two molecules should be in the same diffusion state should be kept in the colocalization analysis. Thus, I encourage that the

authors replace the relevant figure panels (Figs 4d and 4e) with the ones they showed in the response letter (Response Letter Figure 12) and update the text (main text and methods in accordance).

We thank the Reviewer #2 for the valuable comment. By following the comment, we replaced the figures regarding the colocalization analysis (Figs. 4d, e and Supplementary Fig. 13) and methods as follows:

Main text (Line 211-215) :

“In the meantime, the lifetime of AT1R-GRK5 complexes was slightly increased (Fig. 4e). The YM pre-treatment reverted the Ang II-induced decrease of association rate to the similar extent of TRV027 stimulation. The TRV027 stimulation statically significantly increased the lifetime of AT1R-GRK5 complexes (Fig. 4e and Supplementary Fig. 13e,f.). These results suggested an enhanced availability of ligand-bound AT1R to active GRK5 in the TRV/Ang II + YM treatments.”

Methods (Line 639-641):

“Here, the colocalization of two particles is detected if the two particles are within 100 nm in the same frame, which corresponds to 2~3 SDs of the total position accuracy, and if the two particles are in the same diffusion state.”

We are afraid that our intention was not correctly conveyed to the Reviewer #2 as to why we had changed the criteria of the colocalization in the previous point-to-point response letter. Firstly, we agree with the Reviewer #2's comment “If AT1R and GRK5 dissociate upon Ang II and now are in different mobility states, then they should no longer considered colocalized”. The criterion “the same diffusion state” probably contributes to reduce the false positive detection of colocalization (co-diffusion). On the other hand, it would risk a change in the detection criteria of co-diffusion if the actual diffusion dynamics differ between the corresponding diffusion states of AT1R and GRK among ligand conditions. Regarding the step-size distribution (Supplementary Fig. 11a, b), the HMM analysis classified AT1R and GRKs into four similar states. However, the MSD-dt plots indicated different confinement lengths between the corresponding diffusion states of AT1R and GRK5 (Supplementary Fig. 11e, f). As we noted in the previous response letter, the ligand-induced changes of the confinement length were reversed between AT1R and GRK5, suggesting that different diffusion dynamics are classified into the corresponding diffusion states. Therefore, we decided to choose the relaxed criteria to avoid a biased detection in the in different ligand conditions.

We should admit, however, that it is challenging to assess how false detection rates and detection biases have changed with the relaxation of criteria, owing to the lack of the ground truth. Since we have confirmed that regardless of the detection methods, the qualitative conclusions remain, we revised the manuscript as described above according to the Reviewer #2's comments.

3. I agree with the other reviewers that the authors could further improve discussions on the molecular mechanisms of the observed transducer bias. The new Figure 7 is helpful, but some critical details are absent. For example, when does AT1R phosphorylation by GRK5/6 take place? Should the Gq-GRK5/6 complexes remain bound after the phosphorylation happens? If so, when do the Gq-GRK5/6 complexes fall of the receptor? After arrestin recruitment, endocytosis, etc.? I am aware that these are not directly studied in this work. If answers to these are unclear at present, then they could be question marks on the corresponding steps in the diagram, so the readers are informed. The authors may also want to update the GRK2/3 part of this diagram since GRK2/3 recruitment follows G β dissociation from G α . In addition, I wondered if the authors have done pERK assays in DGRK cells (they had endocytosis data from these cells). I do not suggest additional experiments, but if they already have any data on that front, they may consider including those to strengthen the signaling aspects of this work.

As recommended by the Reviewer #2, we revised the summary figure (Fig. 7 in the revised manuscript) by reflecting the following three points.

1. *“When does AT1R phosphorylation by GRK5/6 take place?”*

In our single-molecule microscopy experiment, we observed that TRV027 stimulation increased the proportion of both AT1R and GRK5 molecules in the slower (slow and immobile) state (Fig. 4f, g). In addition, both molecules in the immobile state formed oligomers (Supplementary Fig. 12d, e). These observations led us to reasonably assume that GRK5/6 molecules encounter and efficiently phosphorylate AT1R molecules in the microdomain, where both molecules in the immobile state are enriched.

2. *“Should the Gq-GRK5/6 complexes remain bound after the phosphorylation happens? If so, when do the Gq-GRK5/6 complexes fall of the receptor? After arrestin recruitment, endocytosis, etc.?”*

We showed that GRK5/6-mediated β -arrestin recruitment to AT1R occurs to the same extent regardless of Gq activation (Supplementary Fig. 16i). This result suggests that phosphorylation of AT1R is independent of Gq

and can be achieved by GRK5/6 alone (without regulation by Gq). However, we admit that there is another possibility that GRK5/6 phosphorylates Gq-GRK5/6 complexed AT1R, and we believe that a detailed mechanism will be clarified in future studies.

3. *“The authors may also want to update the GRK2/3 part of this diagram since GRK2/3 recruitment follows Gβγ dissociation from Gαq.”*

We have included a model showing that GRK2/3 is recruited to Gβγ, which is dissociated from Gq.

Regarding the pERK level, we have not measured pERK levels in ΔGRK cells. We have shown that pERK responses in the β-arrestin-deficient cells treated with Go6983 was completely abolished (Supplementary Fig. 7a) and that β-arrestin recruitment to AT1R was totally GRK-dependent upon Ang II / Ang II + YM / TRV027 treatments. From these data, we consider that pERK response will be minimally induced in ΔGRK cells because of the lack of following β-arrestin recruitment event as shown experimentally.

4. *There are a number of places in the main text where I find the statement confusing. Please consider revising these.*

For example, in lines 163-166, it reads ‘We note that a large portion of TRV027-induced pERK was sensitive to Go6983 (Supplementary Fig. 7b), indicating a residual Gq-PKC-mediated effect by the biased ligand. Whereas Ang II stimulation induced a robust pERK signal, pERK response was marginal by TRV027 stimulation and nearly undetectable by YM pretreatment (Fig. 3d and Supplementary Fig. 7b-d).’. It would be much easier to follow if the two sentences are swapped in order; even better if the new 2nd sentence reads something like this: ‘we note that the low level of pERK induced by TRV207 was largely diminished upon Go6983 treatment, indicative of a residual Gq-PKC-mediated signaling component.’

We have revised the statements as below in the revised manuscript (Line 160-163).

“Ang II stimulation induced a robust pERK signal, pERK response was marginal by TRV027 stimulation^{41,42} and nearly undetectable by YM pre-treatment (Fig. 3d and Supplementary Fig. 7b-d). We note that the low level of pERK induced by TRV207 was largely diminished upon Go6983 treatment, indicative of a residual Gq-PKC-mediated signaling component (Supplementary Fig. 7b)”

Lines 169-171, it reads 'Compared with Ang II stimulation, the TRV/Ang II + YM treatments showed decreased levels of Ib30-recognizable β -arrestin1. The TRV/Ang II + YM treatments apparently decreased Ib30 reactivity to β -arrestin1 as compared with Ang II stimulation (Supplementary Fig. 5f, g)'. Are these two sentences redundant?

This Reviewer #2's point is correct. We deleted the former sentence.

In lines 195-196, the authors stated that 'Taken together, these results indicate that the GRK-subtype selectivity arises not directly from structural states of the ligand-bound receptor, but from activation level of Gq and, further ...'. There is a gap in the logic flow here (or I missed it somewhere else in the text), which is that YM-pretreatment altered arrestin binding modes on the same Ang II-bound AT1R presumably without altering the structural states of the receptor (since it is the same ligand).

We have rewritten the sentence as below (changes are underlined) in the revised manuscript (Line 190-192).

"Taken together, these results indicate that the GRK-subtype selectivity, a process that occurs prior to arrestin recruitment, arises not directly from structural states of the ligand-bound receptor, but from activation level of Gq and, further..."

In lines 227-230, the authors wrote 'Collectively, these data support a model in which Gq acts as a negative regulator that causes reduction of GRK5 accessibility to the functional domain and thereby results in the lowered association rate between AT1R and GRK5 (Fig. 4h), whereas it simply activates GRK2 (Fig. 4i)'. By 'Gq' I think the authors meant 'active Gq'? Also, I am not sure where the authors inferred 'GRK5 accessibility' to the functional domain (of the receptor presumably)? This sentence could be written as '... in which activation of Gq promotes dissociation of GRK5 from the AT1R and, in the meantime, upregulates GRK2/3 kinase activity via the resulting G $\beta\gamma$ and frees up the receptor for phosphorylation by the activated GRK2/3.'

Thank you for the suggestion. We have rewritten the sentence as below in the revised manuscript (Line 226-229).

"Collectively, these data support a model in which activation of Gq promotes dissociation of GRK5 from the AT1R at the immobile domain (Fig. 4h). In the meantime, Gq activation triggers GRK2 recruitment to the immobile domain (Fig. 4i) presumably via the resulting free G $\beta\gamma$, and frees up the receptor for phosphorylation by the activated GRK2/3."

Reviewer #3 (Remarks to the Author):

The authors have adequately addressed most of the issues I noted during the previous review. A couple of follow-up comments are below (numbered as in the authors' rebuttal).

1)

a. The authors state in the text that “the GRK5/6-dependent β -arrestin-recruitment response is inversely correlated with Gq activity.” However, the key point they show in the main text Fig 3i is the positive correlation for the Δ GRK5/6 cells (i.e., the β -arrestin-recruitment response that is dependent on GRK2/3). Likewise, in the rebuttal the authors state, “In the plots with five ligands, β -arrestin recruitment in Δ GRK5/6 cells is highly correlated with Gq signaling activity ($R^2 > 0.60$), while the correlation with Gi and G12 signaling activity is weak ($R^2 < 0.21$) (Fig. 1j). These results clearly show that there is an inverse correlation between GRK5/6-mediated β -arrestin recruitment and Gq, but such correlation is subtle for Gi and G12 activity.” Since the Δ GRK5/6 data reflects GRK2/3-dependent effects, and the 5 ligand plot of the Δ GRK2/3 (GRK5/6-dependent) data shows only a weak correlation ($r^2 = 0.41/0.19$), how does this support the authors' conclusion? Please clarify.

It seems that there was a confusion of interpretation of the graph. In the correlation plot figure (Fig. 1j and Supplementary Fig. 3i, j), the y-axis denotes GRK dependency as denoted by percentage of arrestin recruitment in the Δ GRK5/6 cells in comparison with the Parent cells (100%). We emphasize that the y-axis is not actual arrestin recruitment response (fold over basal) in Δ GRK5/6 cells. Thus, a smaller value indicates a higher contribution of GRK5/6 to arrestin recruitment to AT1R. To avoid such confusion, we changed the y-axis label as “GRK dependency (% β -arrestin recruitment response in Δ GRK of Parent)” in the revised figure (Fig. 1j and Supplementary Fig. 3i, j). Regarding the smaller correlation coefficient values in the Δ GRK2/3 data than the Δ GRK5/6 data, we speculate that variable GRK5/6 expression levels (Supplementary Fig. 2a-d) in Δ GRK2/3 masked correlation. We consider that this difference suggests existence of another, probably Gq-independent mechanism regulating GRK5/6 activity by the biased ligands, which will require a future in-depth study.

b. The authors exclude AngII from the analysis in the main text since it skews the analysis. However, with AngII excluded, it still appears that a single ligand (SVA) is driving the correlation.

In Response Figure 1, we show the correlation by excluding the SVA ligand. Specifically, the figure shows

scattered plots between β -arrestin recruitment activity and G-protein signaling activity for the four AT1R ligands (SI, SII, Ang(1-7) and TRV027, without Ang II and SVA). As compared with the scattered plots with five ligands (Fig. 1j; $R^2 = 0.64 - 0.69$) or six ligands (Supplementary Fig. 3j, $R^2 = 0.98 - 0.99$), β -arrestin recruitment in Δ GRK5/6 cells is highly correlated with Gq signaling activity ($R^2 > 0.90$).

Although the correlation coefficient is higher in the plots with SVA than without SVA, we believe that the plots with the five ligands, by excluding the outlier Ang II plot, provides the best way to present the correlation data as to avoid selection.

Response Fig.1 Comparison of the β -arrestin recruitment responses in Δ GRK5/6 cells and G-protein-coupling activity related to Fig. 1j.

The scattered plots represent data from the four ligands except for Ang II and SVA (see also Fig. 1j for the plots with all five ligands including SVA and Supplementary Fig. 3i, j for plots with all of the cell lines and the six ligands).

c. Please specify the normalizations in the new panel Fig. 1j (e.g., % AngII max).

As suggested by the Reviewer #3, we have added the explanation in the vertical axis (Fig. 1j in the revised manuscript).

3)

a. *“Normalization to the β -arrestin-recruitment responses . . .” (Ln 171-172): It appears that these data are normalized to the AngII response in each assay, not to the β -arrestin-recruitment response.*

In Fig 7h in the original manuscript, we certainly divided the IB30 recruitment responses into the β -arrestin recruitment response upon Ang II / Ang II + YM / TRV027 treatments, while in Fig. 1e and Supplementary Fig. 7f in the original manuscript, the β -arrestin and the IB30 recruitment responses were normalized to the those of Ang II response, respectively. Therefore, we did not change the sentence in the revised manuscript.

b. *A clearer statement would be something to the effect of:*

“The TRV/Ang II + YM treatments apparently decreased Ib30 reactivity to β -arrestin1 as compared with Ang II stimulation (Supplementary Fig. 5f, g). Normalization of the Ib30 and β -arrestin-recruitment responses showed that Ib30 recruitment is proportional to the β -arrestin1 recruitment levels (Supplementary Fig. 5h). Thus we conclude that the Ib30-recognizable active β -arrestin conformation is similarly induced in the TRV/Ang II + YM treatments.”

We thank the Reviewer #3 for the suggested paragraph. We have rewritten the statement as below in the revised manuscript (Line 166-170).

“The TRV/Ang II + YM treatments apparently decreased Ib30 reactivity to β -arrestin1 as compared with Ang II stimulation (Supplementary Fig. 5f, g). Normalization of the Ib30 and β -arrestin-recruitment responses showed that Ib30 recruitment is proportional to the β -arrestin1 recruitment levels (Supplementary Fig. 5h). Thus we conclude that the Ib30-recognizable active β -arrestin conformation is similarly induced in the TRV/Ang II + YM treatments.”

c. *Please correct the Extended Fig. references here (now Extended Fig 7 rather than Extended Fig 5).*

We have revised the reference (Line 166-170).